# Target-Agnostic Calibration under Distribution Shift with Frequency-Aware Gradient Rectification

Yilin Zhang [1]  Cai Xu [1]  You Wu [1]  Ziyu Guan [1]  Wei Zhao [1]

## Abstract

Real-world model deployments inevitably encounter distribution shifts, rendering the confidence estimates of deep neural networks highly unreliable, posing severe risks in safety-critical applications. Existing methods improve calibration via training-time regularization or post-hoc adjustment, but often rely on access to (or simulation of) target domains, limiting practicality. We propose **F**requency-aware **G**radient **R**ectification (**FGR**), a target-agnostic training framework for robust calibration. From a frequency perspective, FGR applies low-pass filtering to a subset of training images to diminish spurious high-frequency cues and encourage the learning of domain-invariant features. However, the associated information loss can degrade In-Distribution (ID) calibration. To resolve this trade-off, FGR treats ID calibration as a hard constraint and rectifies conflicting parameter updates via geometric projection. This ensures a first-order non-increase in the ID calibration objective without introducing an additional loss-balancing coefficient. Extensive experiments on synthetic, real-world, and semantic shift datasets demonstrate that FGR significantly improves calibration under diverse shifts while preserving ID performance, and it remains compatible with post-hoc calibration methods. Our code is available at https://github.com/YilinZhang107/FGR-Calib.

## 1. Introduction

Overconfident errors can lead to catastrophic consequences for artificial intelligence models. Therefore, it is equally critical for deployed models to provide reliable confidence

[1]School of Computer Science and Technology, Xidian University, Xi'an, China. Correspondence to: Wei Zhao <ywzhao@mail.xidian.edu.cn>.

*Proceedings of the 43rd International Conference on Machine Learning*, Seoul, South Korea. PMLR 306, 2026. Copyright 2026 by the author(s).

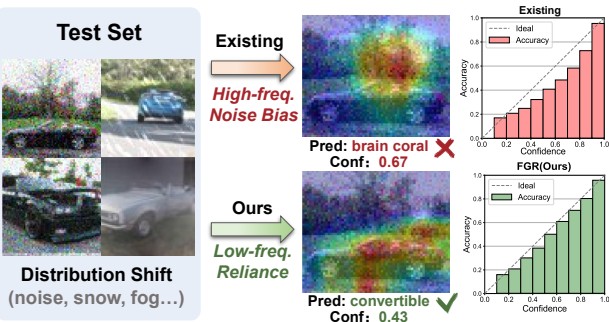

*Figure 1.* Comparison of Grad-CAM heatmaps and reliability diagrams under distribution shift. Existing methods produce overconfident incorrect predictions based on spurious noisy patterns, whereas our method relies on domain-invariant features and provides more reliable confidence estimates.

estimates alongside accurate predictions, especially in high-stakes applications such as autonomous driving (Cao et al., 2024) and clinical diagnostics (Penso et al., 2024). Calibration quantifies the alignment between predicted confidence and true accuracy, e.g., among predictions with 0.6 confidence, approximately 60% should be correct.

In practical deployment, models inevitably encounter distribution shifts, where test inputs differ from the training data due to common factors (as shown in the left part of Figure 1) like weather changes, lighting variations, and sensor inconsistencies (Ovadia et al., 2019; Naganuma et al., 2025). These universal shifts make calibration exceptionally difficult: models often fail to generalize their confidence estimates, leading to dangerous overconfidence on shifted data. Consequently, maintaining robust calibration under diverse and unforeseen shifts remains a significant challenge.

Existing approaches addressing this issue can be broadly divided into Target-Domain-Aware and Target-Domain-Agnostic methods. Target-Domain-Aware methods leverage information from target domains or domain generation rules to learn adaptive calibration strategies (Tomani et al., 2021; Yu et al., 2022). These approaches either utilize known multi-domain data to train input-specific calibration functions, or construct validation sets that simulate target domain characteristics for temperature scaling (Wang et al., 2024). However, because these methods depend on explicit or simu-

lated target domain information, their practical applicability in real-world scenarios is limited, particularly when faced with diverse and unknown distribution shifts.

In contrast, Target-Domain-Agnostic methods do not require access to target domain information, instead implicitly suppress overconfidence through training-time modifications. Key paradigms include: (1) Calibration-aware losses. For example, Focal Loss reweights easy examples (Mukhoti et al., 2020), and MaxEnt Loss encourages predictions to stay close to the statistical patterns observed during training, reducing overreaction to unseen shifts (Neo et al., 2024). (2) Regularization techniques like label smoothing (Müller et al., 2019) and Mixup (Thulasidasan et al., 2019), which prevent overly sharp predictive distributions. While these methods can mitigate miscalibration, they lack explicit mechanisms to handle distribution shifts, often providing only indirect benefits in such scenarios.

To solve this problem, we revisit the reasons for overconfidence on shifted data: distribution shifts predominantly alter high-frequency visual patterns, which deep models often exploit as shortcut cues to form overly sharp predictive distributions (Geirhos et al., 2020). Therefore, relying on these unstable cues often leads to overconfident mispredictions—for instance, misclassifying vehicles based on image noise rather than their invariant shape. Motivated by this, we apply DCT-based low-pass filtering to suppress high-frequency cues, encouraging the model to rely on shape-related information that remains consistent across distributions. As illustrated in Figure 1, this guides the model to focus on robust semantic features rather than spurious texture patterns. However, due to information loss, filtering may also disrupt the fine-grained decision boundaries needed for ID performance, leading to underconfidence.

To resolve this critical trade-off, we propose a training-time gradient rectification strategy that treats ID calibration as a hard constraint during optimization. Specifically, we train the model on a hybrid input set combining original and filtered images. In each step, we compute the geometric relationship between the main loss gradient (e.g., Focal Loss) on the hybrid batch and the ID calibration loss gradient (e.g., Soft-ECE (Karandikar et al., 2021)). When these gradients conflict, we project the main gradient onto the hyperplane orthogonal to the calibration gradient, ensuring under a first-order approximation that the update does not increase the ID calibration loss. This weight-free projection step successfully balances calibration improvements under distribution shifts with strong ID performance. Overall, our contributions are summarized as follows:

- Building on the insight that distribution shifts primarily alter high-frequency visual patterns, we introduce DCT-based low-pass filtering to suppress spurious shortcuts and encourage domain-invariant feature, improving

calibration under shift without target domain access.

- We propose FGR to resolve the trade-off between shift robustness and ID performance by treating ID calibration as a hard constraint and projecting conflict gradients during optimization, with theoretical guarantees.

- Extensive experiments on various shift datasets show FGR substantially improves calibration (reducing ACE by 40% on Camelyon17) while preserving ID performance and compatibility with post-hoc methods.

## 2. Related Work

**Uncertainty Calibration.** Approaches to improve the calibration are typically categorized into post-hoc and training-time methods. Post-hoc methods are applied to pre-trained models without altering their weights. A widely used baseline is Temperature Scaling (TS) (Guo et al., 2017), which optimizes a single scalar to rescale logits. More flexible alternatives include $\rho$-Norm scaling (Zhang & Xie, 2025), which generalizes TS via norm-based adjustments, and isotonic regression (Zadrozny & Elkan, 2001), a non-parametric approach that fits a monotonic mapping between predicted confidence and empirical accuracy. (Tao et al., 2025) clips the feature magnitudes of overconfident samples to increase their predictive entropy. Training-time methods aim to learn calibrated models directly by modifying the loss function or optimization process. Representative approaches include MMCE (Kumar et al., 2018), AvUC (Krishnan & Tickoo, 2020), Soft-ECE (Karandikar et al., 2021) and Dual Focal Loss (Tao et al., 2023b), which directly penalize miscalibration or down-weight confident prediction. Recent evidential methods estimate uncertainty via Dirichlet-distributed class evidence (Chen et al., 2026). Lin et al. (2025) modulate the updated scaling gradient using the uncertainty of each sample. In addition, regularization techniques such as Label Smoothing (Müller et al., 2019), Mixup (Zhang et al., 2021) and CutMix (Yun et al., 2019) can provide indirect calibration benefits.

**Calibration under Distribution Shift.** While the aforementioned methods perform well in-distribution, their calibration performance is often fragile to distribution shifts (Ovadia et al., 2019). To address this, adaptive temperature scaling methods (Yu et al., 2022; Wang et al., 2024; Choi et al., 2024) train regressors using augmented validation sets or known auxiliary domains to estimate input-specific temperature. Other methods incorporate feature density (Tomani et al., 2023), Bayesian inference (Seligmann et al., 2023), or prior training states (Tao et al., 2023a), or test-time distribution estimation for vision-language models (Han et al., 2025) to enhance calibration under shift, often at the cost of additional computation or assumptions. Data augmentation can also improve robustness by exposing the model

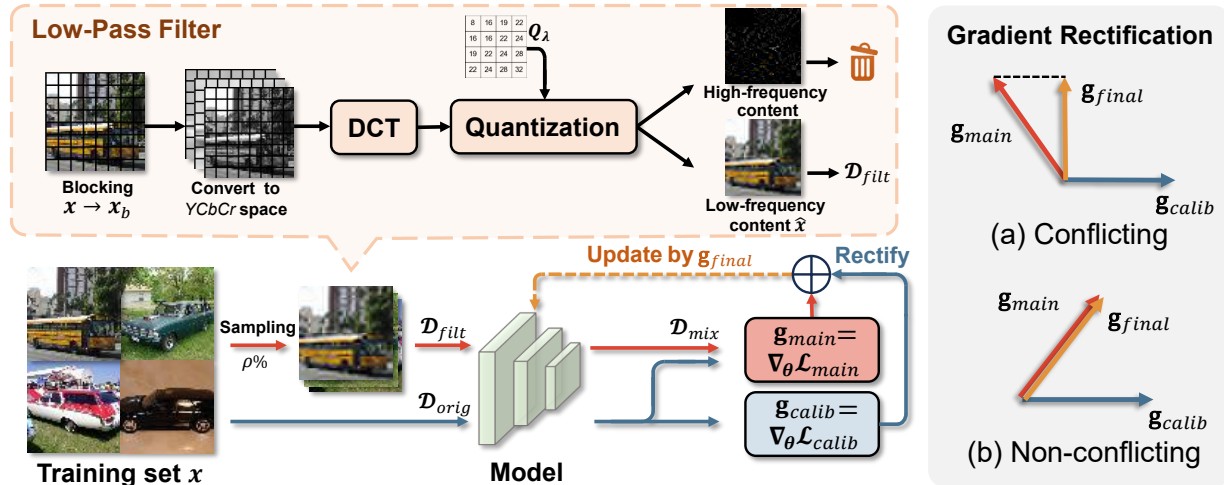

*Figure 2.* Overview of the proposed method. We apply DCT-based low-pass filtering to a subset of training data to suppress high-frequency features and construct a hybrid dataset $\mathcal{D}_{\text{mix}}$. During optimization, we compute gradient $\mathbf{g}_{\text{main}}$ on $\mathcal{D}_{\text{mix}}$ and $\mathbf{g}_{\text{calib}}$ on $\mathcal{D}_{\text{orig}}$, and apply gradient rectification when they conflict to prevent degradation of ID calibration.

to varied inputs (Hendrycks et al., 2020). However, these methods often rely on auxiliary domains, model priors, or handcrafted regularizers—limiting their applicability under unknown or dynamic shifts.

**Frequency-Domain Robustness.** Recent frequency-domain studies reveal insights into model robustness (Vaish et al., 2024). Yin et al. (2019) show that models often exploit high-frequency non-robust statistics, while Fridovich-Keil et al. (2022) reveal model sensitivity to spectral characteristics of input images. Li et al. (2023) demonstrate that discarding high-frequency components can preserve semantically meaningful information through DCT. Wang et al. (2023) found that the model relies on specific frequency characteristics in the data as a shortcut for classification. Building upon these insights, our work leverages frequency filtering to build inherent distribution shift calibration robustness without any target information.

## 3. Problem Formulation

We consider a image classification task over a dataset $\mathcal{D} = \{(\boldsymbol{x}_i, y_i)\}_{i=1}^{N}$ with $N$ samples, where $\boldsymbol{x}_i \in \mathcal{X}$ represents the input and $y_i \in \{1, 2, \ldots, K\}$ denotes the ground-truth class label for $K$ classes. A neural network $f(\theta)$ with parameters $\theta$ maps input $\boldsymbol{x}_i$ to logits $\boldsymbol{z}_i = f(\boldsymbol{x}_i; \theta)$. After applying the Softmax function, the predicted probability for class $k$ is given by $p_{ik} = \frac{\exp(z_{ik})}{\sum_{j=1}^{K} \exp(z_{ij})}$. The predicted class label $\hat{y}_i$ and corresponding confidence $\hat{p}_i$ are defined as:

$$\hat{y}_i = \arg\max_k p_{ik}, \ \ \hat{p}_i = \max_k p_{ik}, \ k \in \{1, \ldots, K\}. \ (1)$$

A model is perfectly calibrated if its confidence scores accurately reflect the true likelihood of correctness, formally

satisfying $P(\hat{y} = y | \hat{p} = p) = p$ for all $p \in [0, 1]$. In practice, since the true posterior distribution is unknown, this ideal condition is approximated by partitioning predictions into bins based on confidence levels (Gawlikowski et al., 2023).

**Expected Calibration Error (ECE):** ECE is the most widely-used metric to quantify calibration performance. The confidence interval $[0, 1]$ is partitioned into $M$ bins $\{B_m\}_{m=1}^{M}$, where bin $B_m$ contains all samples with confidence $\hat{p} \in \left(\frac{m-1}{M}, \frac{m}{M}\right]$. Then, ECE is calculated as the weighted average of absolute differences between accuracy and confidence across all bins:

$$\text{ECE} = \sum_{m=1}^{M} \frac{|B_m|}{N} |\text{acc}(B_m) - \text{conf}(B_m)|. \quad (2)$$

where $\text{acc}(B_m)$ and $\text{conf}(B_m)$ represent the empirical accuracy and average confidence within bin $B_m$, respectively. Additional calibration metrics include Classwise ECE (CECE) (Kull et al., 2019), which evaluates calibration separately for each class to identify class-specific miscalibration patterns, and Adaptive Calibration Error (ACE) (Nixon et al., 2019), which employs adaptive binning to mitigate estimation instability in sparse confidence regions. Detailed formulations are provided in Appendix C.1.

**Distribution Shift Calibration:** In real-world deployment, models encounter distribution shifts where test data $\mathcal{D}_{\text{test}}$ differs from training data $\mathcal{D}_{\text{train}}$. Our goal is to learn a model that maintains well-calibrated predictions on both in-distribution data and under various unseen distribution shifts, without requiring access to target domain information during training.

## 4. Method

From a frequency perspective, we first use low-pass filtering to enhance robustness under distribution shift. However, this introduces a trade-off, as filtering can degrade ID calibration by causing underconfidence. We resolve this with a gradient rectification mechanism that treats ID calibration as a hard, first-order constraint, ensuring robustness gains do not compromise ID performance.

### 4.1. Low-Pass Filtering for Robust Features

Prior work has shown that distribution shifts often alter high-frequency visual patterns that models exploit as predictive shortcuts (Fridovich-Keil et al., 2022; Li et al., 2023). This can lead to overconfident predictions based on features with spurious correlations. Motivated by this observation, we apply a low-pass filter to suppress source-specific high-frequency cues, thereby encouraging the model to rely more on domain-invariant semantic features under shift. We specifically choose a Discrete Cosine Transform (DCT)-based approach (Khayam, 2003) for its strong energy compaction property and block-wise processing, which avoids the global ringing artifacts of Fourier-based methods. This makes it highly effective for preserving core semantics while robustly discarding high-frequency noise.

At the beginning of each training epoch, we randomly select a proportion $\rho$ of the training samples and apply the low-pass filtering to obtain a filtered subset $\mathcal{D}_{\text{filt}}$. The remaining $(1-\rho)$ portion of samples are kept unchanged, forming the unfiltered subset $\mathcal{D}_{\text{orig}}$. We then define the hybrid training set as $\mathcal{D}_{\text{mix}} = \mathcal{D}_{\text{filt}} \cup \mathcal{D}_{\text{orig}}$.

For the filtering process, we implement the DCT approach using a block-wise method. Given an input image $\boldsymbol{x} \in \mathbb{R}^{H \times W \times 3}$, we first convert it to the YCbCr color space. Each channel is then partitioned into non-overlapping $8 \times 8$ blocks. This local processing is robust to common texture distortions without introducing global artifacts. We apply a 2D-DCT to each block $\boldsymbol{x}_b$:

$$\mathbf{F}_b = \text{DCT}(\boldsymbol{x}_b), \qquad (3)$$

where $\mathbf{F}_b \in \mathbb{R}^{8 \times 8}$ contains the frequency coefficients. These coefficients are then quantized by dividing element-wise with the given quantization matrix $\mathbf{Q}_\lambda$ and rounding to the nearest integer:

$$\mathbf{F}_b^{(\text{q})} = \text{round}\left(\frac{\mathbf{F}_b}{\mathbf{Q}_\lambda}\right), \qquad (4)$$

where $\lambda \in [1, 100]$ controls the filtering strength. $\mathbf{Q}_\lambda$ is obtained by scaling standard JPEG tables, making the filtering intensity easily adjustable. A lower $\lambda$ corresponds to more aggressive filtering. The quantized coefficients are then de-quantized and inversely transformed to reconstruct

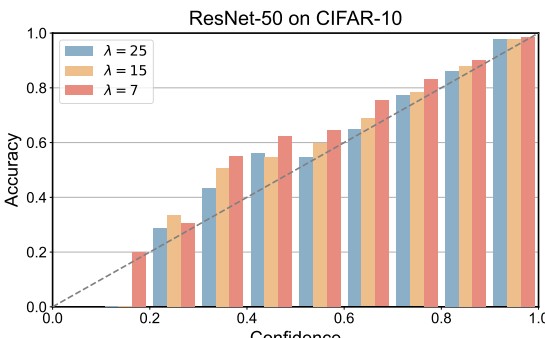

*Figure 3.* Effect of filtering strength $\lambda$ on ID calibration. Lower $\lambda$ (more aggressive filtering) leads to underconfidence.

the filtered block:

$$\hat{\mathbf{F}}_b = \mathbf{F}_b^{(\text{q})} \cdot \mathbf{Q}_\lambda, \quad \hat{\boldsymbol{x}}_b = \text{DCT}^{-1}\left(\hat{\mathbf{F}}_b\right). \qquad (5)$$

The final filtered image $\boldsymbol{x}'$ is formed by reassembling all $\hat{\boldsymbol{x}}_b$ blocks and converting back to the RGB space. This hybrid strategy exposes the model to both original full-spectrum inputs and low-pass filtered versions, discouraging reliance on domain-specific artifacts.

### 4.2. Gradient Rectification

While low-pass filtering encourages the model to learn domain-invariant features and improves robustness under distribution shift, it may simultaneously degrade ID calibration performance. As shown in the reliability diagrams in Figure 3, more aggressive filtering (i.e., a lower $\lambda$ value) removes fine-grained cues, causing the model to become underconfident on ID data. To address this fundamental trade-off, we propose a gradient rectification mechanism that treats ID calibration as a hard constraint during optimization, rather than a competing objective. This ensures that updates aimed at improving shift robustness do not compromise ID calibration. Specifically, our approach manages two potentially conflicting objectives:

- **Generalization Objective:** We aim to learn robust features through a main classification loss $\mathcal{L}_{\text{main}}$ (e.g., Cross Entropy or Focal Loss) computed on the mixed dataset $\mathcal{D}_{\text{mix}}$ containing both original and frequency-filtered images. The gradient for this objective is:

$$\mathbf{g}_{\text{main}} = \nabla_\theta \mathcal{L}_{\text{main}}(\theta; \mathcal{D}_{\text{mix}}). \qquad (6)$$

- **ID Calibration Objective:** We preserve ID calibration through a explicit calibration loss $\mathcal{L}_{\text{calib}}$ (e.g., Soft-ECE (Karandikar et al., 2021)) computed exclusively on original training data $\mathcal{D}_{\text{orig}}$. The objective's gradient is:

$$\mathbf{g}_{\text{calib}} = \nabla_\theta \mathcal{L}_{\text{calib}}(\theta; \mathcal{D}_{\text{orig}}). \qquad (7)$$

When these two gradients conflict (i.e., point in opposite directions), we rectify the main gradient by projecting it onto the hyperplane orthogonal to the calibration gradient. Otherwise, if the gradients are aligned, the update proceeds normally, as illustrated in Figure 2 (right). Formally, the final rectified gradient $\mathbf{g}_{\text{final}}$ is defined as:

$$\mathbf{g}_{\text{final}} = \begin{cases} \mathbf{g}_{\text{main}}, & \text{if } \mathbf{g}_{\text{main}} \cdot \mathbf{g}_{\text{calib}} \geq 0 \\ \mathbf{g}_{\text{main}} - \frac{\mathbf{g}_{\text{main}} \cdot \mathbf{g}_{\text{calib}}}{\|\mathbf{g}_{\text{calib}}\|^2 + \epsilon} \mathbf{g}_{\text{calib}}, & \text{otherwise}, \end{cases}$$
(8)

where $\epsilon$ is a small constant for numerical stability. This projection guarantees a non-increase of the ID calibration loss $\mathcal{L}_{\text{calib}}$ to a first-order approximation, while avoiding any additional loss-balancing coefficient (Yu et al., 2020; Zhu et al., 2023).

In our implementation, we adopt Dual Focal Loss (DFL) (Tao et al., 2023b) as the main objective $\mathcal{L}_{\text{main}}$ over the mixed dataset $\mathcal{D}_{\text{mix}}$. DFL introduces a dual modulation mechanism that penalizes both underconfident and overconfident predictions, yielding better calibration potential than standard cross-entropy. For a sample $\boldsymbol{x}$, its formulation is:

$$\mathcal{L}_{\text{main}} = -\sum_{k=1}^{K} y_k \left(1 - \hat{p}_k(\boldsymbol{x}) + \hat{p}_j(\boldsymbol{x})\right)^\gamma \log \hat{p}_k(\boldsymbol{x}), \quad (9)$$

where $j$ denotes the highest-scoring incorrect class and $\gamma$ is a tunable focusing parameter.

To supervise ID calibration, we employ Soft-Binned ECE (Soft-ECE) (Karandikar et al., 2021) as the calibration loss $\mathcal{L}_{\text{calib}}$, computed on unfiltered samples $\mathcal{D}_{\text{orig}}$. Soft-ECE provides a differentiable approximation of the standard ECE by using a soft, temperature-controlled binning function. Its general form is:

$$\mathcal{L}_{\text{calib}} = \left( \sum_{m=1}^{M} \frac{|S_m|}{N} |\text{acc}(S_m) - \text{conf}(S_m)|^2 \right)^{1/2}, \quad (10)$$

where $S_m$ are soft bins derived from a membership function. A detailed description of the Soft-ECE formulation and its implementation is provided in Appendix C.4.2.

**Projection Property.** The rectification step is asymmetric and should not be interpreted as exactly minimizing a scalarized sum of the main and calibration losses. Instead, FGR performs a one-sided constrained update: it follows the main-loss gradient on $\mathcal{D}_{\text{mix}}$ whenever this direction is compatible with ID calibration, and otherwise applies the minimum correction needed to avoid increasing the ID calibration loss to first order.

**Proposition 4.1** (Projection-based ID calibration preservation)**.** *Assume $\mathcal{L}_{calib}$ is differentiable and $\mathbf{g}_{calib} \neq \mathbf{0}$. Let*

$$\mathcal{C}_{ID} = \{\mathbf{g} \mid \mathbf{g}^\top \mathbf{g}_{calib} \geq 0\}$$

*be the set of update directions that do not increase the ID calibration objective to first order. Ignoring the small numerical stabilizer $\epsilon$, the FGR direction in Eq. (8) is the Euclidean projection of $\mathbf{g}_{main}$ onto this half-space:*

$$\mathbf{g}_{FGR} = \arg \min_{\mathbf{g} \in \mathcal{C}_{ID}} \|\mathbf{g} - \mathbf{g}_{main}\|_2^2. \quad (11)$$

*Consequently, for a sufficiently small step size $\eta$,*

$$\mathcal{L}_{calib}(\theta - \eta \mathbf{g}_{FGR}) \leq \mathcal{L}_{calib}(\theta) + \mathcal{O}(\eta^2). \quad (12)$$

Proposition 4.1 is the exact optimizer-level statement used in our analysis. It also clarifies the asymmetric nature of FGR compared with standard multi-objective gradient methods: robustness and ID calibration are not treated as two symmetric objectives with a tunable trade-off weight; rather, ID calibration acts as a hard first-order constraint on the robustness-oriented update. A surrogate joint-risk view can still provide high-level intuition, but it is not an equivalence to the implemented algorithm. The proof is given in Appendix E.

## 5. Experiments

We evaluate our method on both synthetic and real-world distribution shift benchmarks. Our experiments demonstrate: (1) significant calibration improvements under distribution shift while maintaining competitive accuracy; (2) its ability to maintain strong calibration performance on clean in-distribution data; (3) the necessity of both frequency filtering and gradient rectification components; (4) gradient conflict analysis reveals how our rectification mechanism prevents overconfidence during later training phases; and (5) improved focus on semantically meaningful features.

### 5.1. Experimental Setup

**Datasets.** We evaluate our method on a diverse set of benchmarks. For **synthetic shifts**, we use **CIFAR-10/100** and **Tiny-ImageNet** as in-distribution (ID) data, with their corresponding corrupted versions (**CIFAR-10/100-C**, **Tiny-ImageNet-C**) (Hendrycks & Dietterich, 2019) serving as distribution shift test sets. These corrupted datasets cover 15 common corruption types across 5 severity levels. For **real-world shifts**, we use **Camelyon17**, **iWildCam**, and **FMoW** from the **WILDS** benchmark (Koh et al., 2021), which feature naturally occurring domain shifts from different hospitals, camera traps, and geographic regions, respectively. Additionally, we evaluate on **Office-Home** (Venkateswara et al., 2017), a multi-domain benchmark across four visually distinct domains for semantic shift evaluation.

**Compared Methods.** We compare our training-time method against a suite of strong baselines: standard Cross-Entropy (CE), Label Smoothing (LS-0.05) (Müller et al.,

*Table 1.* Test scores (%) of different methods on synthetic (top) and real-world (bottom) distribution shift test sets. For synthetic datasets, results are averaged over 15 corruption types across 5 severity levels. The "w/ TS" columns show ECE values with temperature scaling post-hoc calibration. The best scores are highlighted in **bold**.

| | **CIFAR-10-C** / DenseNet-121 | | | | | **CIFAR-100-C** / DenseNet-121 | | | | | **Tiny ImageNet-C** / DenseNet-121 | | | | |
| Loss Fn. | Acc. | ECE | w/ TS | CECE | ACE | Acc. | ECE | w/ TS | CECE | ACE | Acc. | ECE | w/ TS | CECE | ACE |
|---|---|---|---|---|---|---|---|---|---|---|---|---|---|---|---|
| CE | 74.05 | 23.51 | 17.16 | 4.87 | 4.38 | 48.44 | 41.21 | 13.31 | 0.91 | 0.74 | 24.27 | 25.83 | 36.23 | 0.44 | 0.43 |
| LS-0.05 | 73.65 | 16.42 | 18.36 | 3.81 | 3.80 | 51.37 | 19.47 | 16.20 | 0.55 | 0.51 | 25.96 | 16.11 | 16.11 | **0.37** | 0.39 |
| Mixup | 75.41 | 14.80 | 17.81 | 3.79 | 3.78 | 53.38 | 12.43 | 16.32 | 0.50 | 0.53 | 26.26 | **11.75** | 16.77 | **0.37** | 0.40 |
| AugMix | 82.49 | 11.03 | **8.51** | 3.31 | 3.24 | 59.87 | 16.14 | 7.84 | 0.50 | 0.49 | 14.51 | 22.55 | 11.98 | 0.48 | 0.50 |
| FLSD-53 | 72.61 | 13.58 | 14.99 | 3.74 | 3.70 | 49.39 | 13.74 | 10.04 | 0.56 | 0.53 | 22.30 | 15.35 | 47.58 | 0.41 | 0.42 |
| DFL | 70.18 | 16.19 | 15.12 | 4.28 | 4.23 | 50.17 | 9.99 | 8.82 | 0.51 | 0.49 | 23.84 | 13.12 | 16.62 | 0.38 | 0.39 |
| MaxEnt M | 71.98 | 11.62 | 13.63 | 3.62 | 3.62 | 48.34 | 11.05 | 10.38 | 0.57 | 0.54 | 21.14 | 21.79 | 17.05 | 0.46 | 0.46 |
| BSCE-GRA | 72.46 | 11.45 | 14.11 | 3.64 | 3.63 | 49.22 | 11.69 | 10.86 | 0.54 | 0.52 | 21.47 | 13.08 | 19.96 | 0.40 | 0.41 |
| **FGR** | 75.12 | **9.02** | 9.90 | **3.12** | **3.09** | 52.66 | **8.53** | **7.57** | **0.47** | **0.46** | 24.03 | 12.39 | **11.16** | 0.40 | **0.38** |

| | **Camelyon17** / DenseNet-121 | | | | | **iWildCam** / ResNet-50 | | | | | **Fmow** / DenseNet-121 | | | | |
| Loss Fn. | Acc. | ECE | w/ TS | CECE | ACE | Acc. | ECE | w/ TS | CECE | ACE | Acc. | ECE | w/ TS | CECE | ACE |
|---|---|---|---|---|---|---|---|---|---|---|---|---|---|---|---|
| CE | 86.83 | 12.23 | 6.18 | 12.612 | 12.186 | 77.51 | 14.99 | 3.61 | 0.196 | 0.163 | 52.31 | 41.94 | 5.70 | 1.42 | 1.01 |
| LS-0.05 | 85.86 | 8.26 | 7.55 | 13.859 | 13.858 | 77.39 | 8.60 | 5.98 | 0.224 | 0.230 | 50.89 | 33.64 | 6.91 | 1.14 | 0.97 |
| Mixup | 88.33 | 2.62 | 2.05 | 10.097 | 10.105 | 75.85 | 3.50 | 3.88 | 0.164 | 0.159 | 51.53 | 25.16 | 5.53 | 0.98 | 0.76 |
| AugMix | 70.99 | 16.99 | 8.19 | 19.678 | 19.680 | 49.07 | 22.18 | 3.36 | 0.409 | 0.419 | 27.47 | 48.68 | **3.08** | 1.86 | 1.77 |
| FLSD-53 | 87.10 | 6.33 | 3.67 | 11.787 | 11.788 | 74.04 | 7.82 | 7.69 | 0.223 | 0.212 | 53.81 | 28.72 | 4.09 | 1.03 | 0.80 |
| DFL | 88.03 | 2.74 | 2.12 | 9.957 | 9.956 | 73.52 | 6.97 | 3.57 | 0.225 | 0.217 | 53.55 | 26.59 | 4.30 | 0.97 | 0.77 |
| MaxEnt M | 85.67 | 3.96 | 2.71 | 12.930 | 12.932 | 75.11 | 7.14 | 3.08 | 0.196 | 0.182 | 53.04 | 30.69 | 4.85 | 1.09 | 0.85 |
| BSCE-GRA | 86.44 | 5.53 | 2.50 | 11.336 | 11.334 | 75.19 | 5.02 | 3.74 | **0.150** | 0.156 | 53.53 | 27.40 | 3.60 | 0.99 | 0.77 |
| **FGR** | 89.19 | **2.36** | **1.82** | **5.714** | **5.691** | 76.11 | **3.34** | **2.97** | 0.155 | **0.152** | 51.95 | **25.06** | 3.84 | **0.92** | **0.74** |

2019), Mixup (Zhang et al., 2021), AugMix (Hendrycks et al., 2020), FLSD-53 (Mukhoti et al., 2020), Dual Focal Loss (DFL) (Tao et al., 2023b), MaxEnt (Neo et al., 2024), and BSCE-GRA (Lin et al., 2025). We also report results with and without post-hoc Temperature Scaling (TS) (Guo et al., 2017) to evaluate compatibility.

**Implementation Details.** For experiments on CIFAR and Tiny-ImageNet, we follow the setup in (Mukhoti et al., 2020), training ResNet-50/110, DenseNet-121, and Wide-ResNet-26 models from scratch for 350 epochs. For our FGR method, we also train from scratch, but introduce the frequency filtering and gradient rectification starting from epoch 200. This allows the model to first establish robust classification boundaries before applying our calibration-focused optimizations. For WILDS datasets, all methods fine-tune ImageNet pre-trained models following the official WILDS training protocols. FGR adds only 18% training time compared to standard training (see Appendix D.6). Additionally, we provide a two-stage fine-tuning approach that can efficiently improve calibration of existing models with reduced cost. Detailed hyperparameters and fine-tuning configurations are provided in Appendix C.

### 5.2. Performance under Distribution Shift

We first evaluate our method on both synthetic corruption datasets and real-world distribution shift benchmarks. Table 1 summarizes the results across multiple datasets. On synthetic shifts like CIFAR-10-C and CIFAR-100-C, our ap-proach significantly reduces ECE compared to strong baselines like MaxEnt M and BSCE-GRA. This demonstrates that encouraging reliance on domain-invariant features improves robustness. Unlike methods with aggressive augmentation such as AugMix, which perform well on synthetic data but poorly on real-world WILDS datasets, our method also generalizes effectively to real-world shifts. It outperforms all baselines on Camelyon17 in both ECE (2.36%) and CECE (5.714%), and remains competitive on FMoW and iWildCam. The results in Table 1 ("w/ TS" columns) also confirm that our method is compatible with post-hoc temperature scaling, often leading to further calibration improvements.

Furthermore, Figure 4 shows that our method maintains low ECE across all corruption severities, especially in high-shift scenarios. Figure 5 further visualizes the reliability diagrams on iWildCam test set, where our method achieves the best alignment with the ideal diagonal line, indicating superior calibration performance. Most importantly, while significantly improving robustness under shift, FGR preserves—and in some cases enhances—calibration on clean In-Distribution data. Detailed comparisons on ID benchmarks are provided in Appendix D.3.

### 5.3. Robustness on Semantic Shift Datasets

To evaluate robustness beyond synthetic corruptions and natural shifts, we conduct experiments on Office-Home, a semantic shift benchmark featuring four visually distinct do-

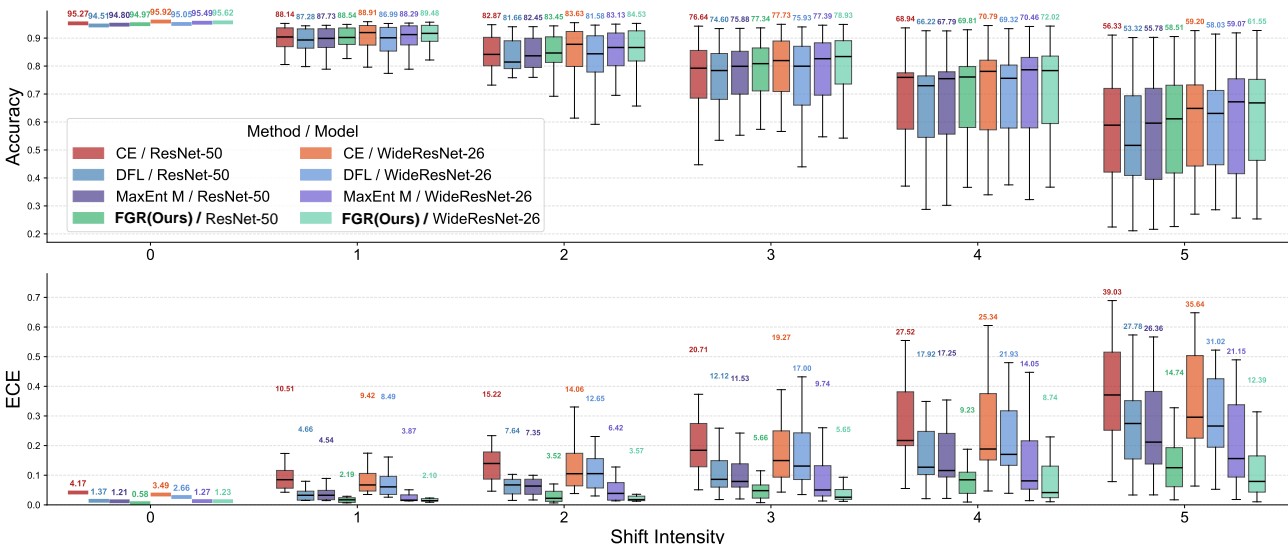

*Figure 4.* Test Accuracy↑ (%) and ECE↓ (%) of different methods trained on CIFAR-10 across ResNet-50 and WideResNet-26. Results are evaluated on clean test data and corrupted test sets with intensity levels 1-5. Each box shows quartiles summarizing results across all 15 corruption types. Numbers above boxes indicate the mean values across all corruption types.

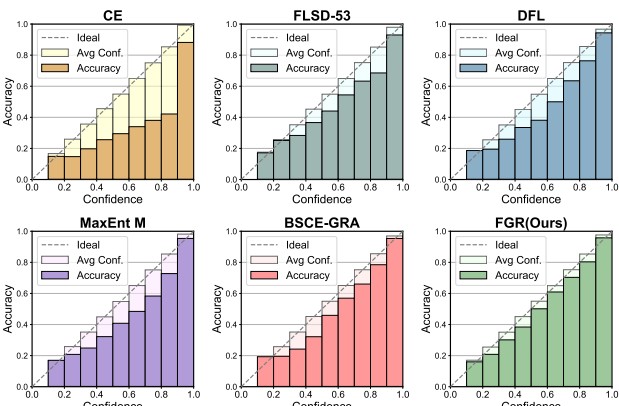

*Figure 5.* Reliability diagrams of six methods on the distribution-shift test set of iWildCam.

*Table 2.* Performance on Office-Home dataset. Results are averaged across four leave-one-domain-out experiments.

| | Method | Acc. ↑ | ECE ↓ | TS-ECE ↓ | CECE ↓ | ACE ↓ |
|---|---|---|---|---|---|---|
| **ID** | CE | 63.22 | 18.63 | 3.12 | 0.858 | 0.527 |
| | FLSD-53 | 63.44 | 6.95 | 3.34 | 0.701 | 0.441 |
| | DFL | 63.06 | 8.93 | 3.21 | 0.702 | 0.457 |
| | BSCE-GRA | 61.81 | 6.56 | 3.76 | 0.699 | 0.448 |
| | **FGR** | 63.19 | **6.32** | **2.96** | **0.682** | **0.438** |
| **OOD** | CE | 34.20 | 36.45 | 15.11 | 1.429 | 1.238 |
| | FLSD-53 | 33.31 | 22.92 | 17.03 | 1.084 | 0.995 |
| | DFL | 34.17 | 22.91 | 14.51 | 1.061 | 0.975 |
| | BSCE-GRA | 32.55 | 21.09 | 15.29 | 1.052 | 0.991 |
| | **FGR** | 34.03 | **20.41** | **13.93** | **1.018** | **0.971** |

mains. Following standard domain generalization protocols with leave-one-domain-out evaluation, we fine-tuned the pre-trained ResNet-50 on three domains and test on the held-out domain. Table 2 reports the average results across four configurations. On both the ID and semantic shift test sets, FGR demonstrated optimal calibration performance across all metrics with negligible accuracy degradation. This indicates that our frequency-based approach not only handles low-level data corruption but also generalizes to high-level semantic distribution shifts.

### 5.4. Ablation Study

We conduct two sets of ablation studies. First, Table 3 isolates the effects of frequency filtering and gradient rectification on ID and synthetic-shift benchmarks. Second,

Table 4 evaluates the same component-level variants on real-world shifts and further compares FGR with PCGrad (Yu et al., 2020) and CAGrad (Liu et al., 2021a), two symmetric multi-objective gradient methods.

The results in Table 3 reveal a clear trade-off. Filter Only can improve calibration under some shifts (e.g., ECE on CIFAR-100-C is 9.70%), but it may substantially degrade ID calibration (ECE on CIFAR-10 is 11.54%). Rect. Only better preserves ID calibration, but does not provide sufficient robustness under shift. Weighted Sum can work after carefully adjusting the loss weight, but its behavior is less consistent across datasets. In contrast, FGR combines filtering with asymmetric rectification, achieving a better calibration trade-off without introducing an additional loss-balancing coefficient.

Table 4 shows that the same component-level trend also appears on real-world shifts. To further examine whether

*Table 3.* Ablation study results on In-Distribution (ID) and distribution shift test sets.

| Method | In-Distribution | | | | | | Distribution Shift | | | | | |
|---|---|---|---|---|---|---|---|---|---|---|---|---|
| | CIFAR-10 | | CIFAR-100 | | Tiny-INet | | CIFAR-10-C | | CIFAR-100-C | | Tiny-INet-C | |
| | Acc.↑ | ECE↓ | Acc.↑ | ECE↓ | Acc.↑ | ECE↓ | Acc.↑ | ECE↓ | Acc.↑ | ECE↓ | Acc.↑ | ECE↓ |
| Filter Only | 95.21 | 11.54 | 77.90 | 11.65 | 64.33 | 2.92 | 75.17 | 9.78 | 51.12 | **9.70** | 22.98 | 18.41 |
| Rect. Only | 95.19 | 2.24 | 78.21 | 2.96 | 64.02 | 7.76 | 74.40 | 18.60 | 51.07 | 27.52 | 22.71 | 26.62 |
| Weighted Sum | 95.03 | 1.72 | 77.82 | **2.35** | 64.28 | 3.02 | 75.37 | 7.93 | 51.15 | 11.80 | 22.64 | 17.26 |
| **FGR** | 94.97 | **0.65** | 78.30 | 2.84 | 64.18 | **2.64** | 75.23 | **6.78** | 50.89 | 10.50 | 23.25 | **15.46** |

*Table 4.* Real-world ablation study on Camelyon17 and iWildCam. The best result in each column is highlighted in bold. PCGrad and CAGrad replace the proposed one-sided rectification rule with standard multi-objective gradient-combination mechanisms.

| Method | Camelyon17 | | | | | iWildCam | | | | |
|---|---|---|---|---|---|---|---|---|---|---|
| | Val Acc.↑ | Val ECE↓ | Test Acc.↑ | Test ECE↓ | Test NLL↓ | Val Acc.↑ | Val ECE↓ | Test Acc.↑ | Test ECE↓ | Test NLL↓ |
| Filter Only | 90.79 | 2.49 | 86.97 | 2.24 | 0.3125 | 62.07 | 10.28 | 73.34 | **2.57** | 1.1709 |
| Rect. Only | **91.29** | 1.73 | 87.12 | 2.57 | 0.3157 | 61.74 | **7.81** | 71.80 | 6.05 | 1.2035 |
| Weighted Sum | 90.43 | 1.37 | 87.29 | 1.66 | 0.3046 | **64.02** | 13.15 | 72.27 | 11.17 | 1.4223 |
| PCGrad | 90.56 | 2.53 | **88.68** | 3.38 | **0.2909** | 59.94 | 10.51 | **77.21** | 2.74 | 1.0835 |
| CAGrad | 90.34 | 1.72 | 87.10 | 1.28 | 0.3165 | 60.39 | 9.48 | 74.92 | 4.51 | 1.1044 |
| **FGR** | 90.90 | **0.92** | 87.37 | **1.14** | 0.3091 | 63.61 | 8.19 | 75.22 | 2.62 | **1.0597** |

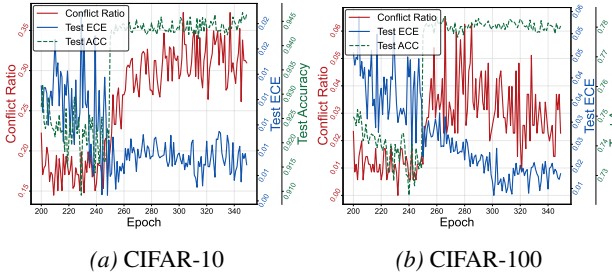

*(a)* CIFAR-10     *(b)* CIFAR-100

*Figure 6.* Evolution of Gradient Conflict Ratio, Test Accuracy, and Test ECE during training. FGR is activated at epoch 200, followed by learning rate decay at epoch 250.

the benefit comes merely from applying a generic gradient-conflict resolution method, we compare FGR with PC-Grad (Yu et al., 2020) and CAGrad (Liu et al., 2021a). In this comparison, PCGrad and CAGrad only replace our asymmetric rectification rule, while the overall training setting is kept unchanged. These symmetric multi-objective methods can obtain higher shifted accuracy in some cases, especially PCGrad on Camelyon17 and iWildCam, but FGR achieves lower test ECE on both datasets. This is consistent with our goal: FGR does not seek a symmetric compromise between robustness and ID calibration, but follows the robustness-oriented update while constraining ID calibration from increasing to first order. The results therefore support the importance of asymmetric rectification for calibration-oriented performance under real-world distribution shifts.

Furthermore, we verify the versatility of FGR. As detailed in Appendix D.4, FGR is compatible with various differentiable calibration losses, consistently yielding performance improvements regardless of the specific objective used.

### 5.5. Gradient Conflict Rate Analysis

To better understand the dynamics of our gradient rectification mechanism, we track the gradient conflict rate throughout training on CIFAR-10/100. Figure 6 visualizes the relationship between conflict rate and test ECE over epochs 200-350. The results reveal two distinct phases. Initially (epochs 200-250), both the gradient conflict rate and ECE decrease rapidly as the model learns to balance the two objectives. However, after the learning rate decay at epoch 250, the conflict rate rises notably. This indicates that with a smaller learning rate, the model tends to sharpen its prediction distributions, which conflicts with maintaining calibration. Crucially, our gradient rectification successfully prevents this sharpening from degrading calibration—despite the increased conflict rate, ECE remains consistently low. This validates that our projection-based rectification effectively constrains the model from becoming overconfident during the later training phase.

### 5.6. Start Epoch Sensitivity

Figure 7 evaluates activating both frequency filtering and gradient rectification from epochs {0, 50, 100, 150, 200, 250} on CIFAR-10/100 and their corrupted counterparts. This experiment specifically studies the start-epoch sensitivity under the from-scratch training protocol used in our main experiments. Starting too early can reduce ID accuracy, since filtering is imposed before the model has learned stable discriminative features. Starting too late leaves too few updates for shifted calibration to benefit. Across these settings, activating FGR in the middle-to-late training stage offers the best balance, which motivates our default start epoch of 200. Additional sensitivity analysis for batch size is provided in Appendix D.5.

*Table 5.* Transformer backbone results on CIFAR-10/100 and Camelyon17. CIFAR-10-C and CIFAR-100-C results are averaged over all corruption types and severities. For Camelyon17, ID denotes the validation split and OOD denotes the shifted test split.

| Model | Method | CIFAR-10 | | CIFAR-10-C | | CIFAR-100 | | CIFAR-100-C | | Camelyon17-ID | | Camelyon17-OOD | |
|---|---|---|---|---|---|---|---|---|---|---|---|---|---|
| | | Acc. | ECE | Acc. | ECE | Acc. | ECE | Acc. | ECE | Acc. | ECE | Acc. | ECE |
| ViT-Small | CE | **98.53** | 0.75 | 90.19 | 5.47 | **90.77** | 3.25 | 73.94 | 9.76 | **92.31** | **6.71** | 82.37 | 16.27 |
| | DFL | 98.48 | 0.69 | 90.37 | **2.62** | 90.24 | 1.39 | 73.40 | **4.26** | 80.94 | 10.90 | 72.93 | **9.15** |
| | **FGR** | 98.37 | **0.42** | 90.79 | 2.80 | 90.40 | **0.93** | **74.15** | 4.74 | 84.58 | 7.81 | 75.99 | 10.14 |
| Swin-Tiny | CE | **97.87** | 0.97 | 84.62 | 8.38 | **87.77** | 2.87 | 64.43 | 14.15 | **94.18** | **4.74** | 82.73 | 15.96 |
| | DFL | 97.85 | 0.82 | 85.23 | 5.21 | 87.27 | 0.99 | 63.73 | 9.54 | 82.65 | 7.27 | 87.17 | 9.54 |
| | **FGR** | 97.86 | **0.46** | 86.39 | 4.79 | 87.12 | **0.80** | 65.39 | 8.18 | 90.68 | 6.16 | **91.46** | **8.21** |
| DeiT-Small | CE | 97.88 | 1.12 | 89.12 | 5.79 | **87.67** | 3.05 | 68.35 | 10.49 | **88.08** | 10.66 | 80.52 | 18.32 |
| | DFL | 97.91 | 2.08 | **89.68** | 2.99 | 87.37 | 4.61 | 68.37 | 5.44 | 87.01 | 8.03 | 73.85 | 11.74 |
| | **FGR** | **98.02** | **0.76** | 89.31 | **2.98** | 87.29 | **2.21** | **68.80** | **4.96** | 85.57 | **5.06** | 82.29 | **8.95** |

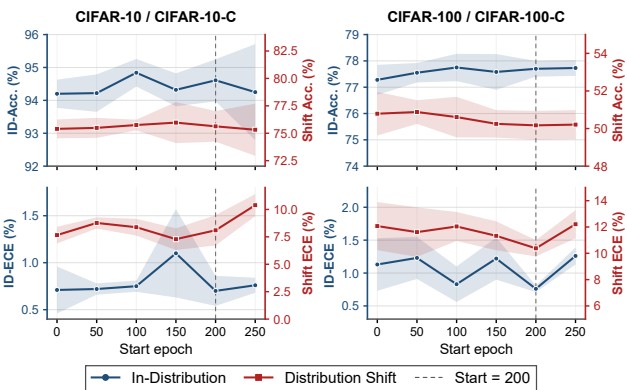

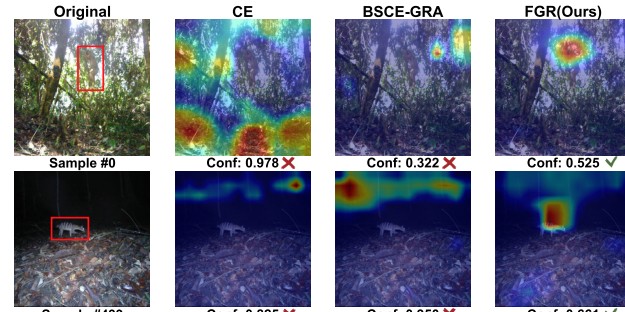

*Figure 8.* Grad-CAM visualization on iWildCam test set. We compare attention maps of CE, BSCE-GRA, and our method.

*Figure 7.* Sensitivity to the epoch at which FGR is activated. We start frequency filtering and gradient rectification at different epochs, and report In-Distribution accuracy and Distribution Shifted ECE on CIFAR-10/100 and CIFAR-10/100-C.

### 5.7. Applicability for Transformer Backbones

Table 5 evaluates whether FGR also transfers to transformer-based architectures, including ViT (Dosovitskiy et al., 2021), Swin Transformer (Liu et al., 2021b), and DeiT (Touvron et al., 2021). We initialize all backbones from ImageNet-pretrained checkpoints, and perform full-parameter fine-tuning on each target training set. On CIFAR-10/100, FGR consistently reduces ID ECE for all three backbones while keeping accuracy comparable to CE and DFL. Under synthetic corruptions, FGR obtains the best or near-best shifted ECE and often improves corrupted accuracy, especially on Swin-Tiny and DeiT-Small. On Camelyon17, FGR also improves OOD calibration over CE for all three architectures and gives the lowest OOD ECE on Swin-Tiny and DeiT-Small. These results suggest that the proposed filtering and rectification mechanisms are not tied to convolutional inductive biases and remain effective for transformer backbones.

### 5.8. Visualization of Model Focus

To understand how our method improves calibration, we visualize model attention patterns on iWildCam using Grad-CAM (Selvaraju et al., 2017). Figure 8 compares attention maps across CE, BSCE-GRA, and our method. The visualizations reveal that baseline methods (CE and BSCE-GRA) often attend to background regions or irrelevant patterns, leading to misclassifications or overconfident incorrect predictions. In contrast, our method consistently focuses on semantically meaningful regions of the target objects, resulting in both improved accuracy and better-calibrated confidence estimates. This confirms that our frequency filtering successfully guides the model to rely on domain-invariant features rather than spurious high-frequency patterns.

### 6. Conclusion

In this paper, we presented Frequency-aware Gradient Rectification (FGR), a target-agnostic training framework designed to mitigate calibration degradation under distribution shift without requiring target domain data. Our approach integrates two complementary mechanisms: frequency-domain filtering, which suppresses spurious high-frequency patterns to encourage reliance on domain-invariant features, and gradient rectification, which treats in-distribution calibration as a hard constraint via geometric projection. Empirical results confirm that this weight-free rectification strategy effectively resolves the fundamental trade-off between shift robustness and ID performance, offering a practical solution for trustworthy model deployment.

## Acknowledgments

This work was supported in part by the National Natural Science Foundation of China under Grants 62425605, 62133012, 62572375, 62303366 and 62472340, in part by the Key Research and Development Program of Shaanxi under Grant 2022ZDLGY01-10, in part by Xidian University Specially Funded Project for Interdisciplinary Exploration under Grant TZJHF202506, and in part by the Fundamental Research Funds for the Central Universities under Grant QTZX25112.

## Impact Statement

This work aims to improve the reliability of confidence estimates in deep learning models under distribution shift, with potential positive impacts in safety-critical applications such as autonomous driving and medical diagnosis. By enabling models to provide well-calibrated predictions without requiring target domain information, our method can help reduce overconfident errors that may lead to harmful decisions in real-world deployments. Additionally, like any calibration technique, improved confidence estimates should be interpreted as one component of trustworthy AI systems, not as a complete solution to model reliability.

Besides, this paper does not raise any ethical concerns. This study does not involve any human subjects, practices to data set releases, potentially harmful insights, methodologies and applications, potential conflicts of interest and sponsorship, discrimination/bias/fairness concerns, privacy and security issues, legal compliance, and research integrity issues.

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

## A. Appendix.

This appendix provides comprehensive supplementary materials for our paper. We organize the content as follows:

- **Section B** presents the complete pseudocode for our Frequency-aware Gradient Rectification (FGR) method.

- **Section C** details the full experimental setup. This includes evaluation metrics (Section C.1), dataset descriptions and preprocessing steps (Section C.2), model architectures and training configurations (Section C.3), and implementation details for both our method and all baselines (Section C.4).

- **Section D** reports additional experimental results, including hyperparameter sensitivity analysis, a comparison of training strategies, and a computational complexity analysis.

- **Section E** provides the proof of the projection property used by FGR.

- **Section F** discusses the limitations and scope of our method.

## B. Overall Algorithm Description

Algorithm 1 summarizes our complete training procedure. The process begins with a warm-up phase, where the model is trained using only the main loss. This is followed by the FGR phase, where in each training step, we sample batches from both the mixed and original datasets to compute the main and calibration gradients, respectively, and apply gradient rectification when their directions conflict.

---

**Algorithm 1** Frequency-aware Gradient Rectification (FGR)

**Input:** Training set $\mathcal{D}$, filtering ratio $\rho$, main loss focusing parameter $\gamma$, FGR start epoch $E_{\text{start}}$
**Output**: Trained model parameters $\theta$

1: Initialize model parameters $\theta$
2: **for** epoch = 1 to Total Epochs **do**
3:     **if** epoch $< E_{\text{start}}$ **then**
4:         // Standard training phase (warm-up)
5:         **for** each batch $b$ from $\mathcal{D}$ **do**
6:             Compute main loss gradient $\mathbf{g}_{\text{main}} = \nabla_\theta \mathcal{L}_{\text{main}}(\theta; b)$
7:             Update $\theta$ using $\mathbf{g}_{\text{main}}$
8:         **end for**
9:     **else**
10:         // FGR training phase
11:         Randomly sample $\rho$ portion of $\mathcal{D}$ to create $\mathcal{D}_{\text{filt}}$ via DCT filtering
12:         Let $\mathcal{D}_{\text{orig}} = \mathcal{D} \setminus \mathcal{D}_{\text{filt}}$, and $\mathcal{D}_{\text{mix}} = \mathcal{D}_{\text{filt}} \cup \mathcal{D}_{\text{orig}}$
13:         **for** each training step **do**
14:             Sample a batch $b_{\text{mix}}$ from $\mathcal{D}_{\text{mix}}$
15:             Sample a batch $b_{\text{orig}}$ from $\mathcal{D}_{\text{orig}}$
16:             Compute main loss gradient $\mathbf{g}_{\text{main}} = \nabla_\theta \mathcal{L}_{\text{main}}(\theta; b_{\text{mix}})$
17:             Compute calibration loss gradient $\mathbf{g}_{\text{calib}} = \nabla_\theta \mathcal{L}_{\text{calib}}(\theta; b_{\text{orig}})$
18:             **if** $\mathbf{g}_{\text{main}} \cdot \mathbf{g}_{\text{calib}} < 0$ **then**
19:                 $\mathbf{g}_{\text{final}} = \mathbf{g}_{\text{main}} - \frac{\mathbf{g}_{\text{main}} \cdot \mathbf{g}_{\text{calib}}}{\|\mathbf{g}_{\text{calib}}\|^2} \mathbf{g}_{\text{calib}}$
20:             **else**
21:                 $\mathbf{g}_{\text{final}} = \mathbf{g}_{\text{main}}$
22:             **end if**
23:             Update $\theta$ using $\mathbf{g}_{\text{final}}$
24:         **end for**
25:     **end if**
26: **end for**

---

# C. Additional Experimental Details

In this section, we provide a comprehensive overview of our experimental setup, including evaluation metrics, dataset specifics, model architectures, training configurations, and implementation details for both our method and the baselines.

## C.1. Evaluation Metrics

We evaluate model calibration using the following metrics. All results reported in the paper are averaged over three independent runs with different random seeds.

**Expected Calibration Error (ECE):** As defined in the main paper, ECE measures the difference between expected accuracy and expected confidence. We compute ECE using $M = 15$ bins.

**Class-wise ECE (CECE):** To assess calibration on a per-class basis, we also report CECE (Kull et al., 2019), which averages the ECE calculated for each class individually. This can reveal if miscalibration is concentrated in specific classes. The formula is:

$$\text{CECE} = \frac{1}{K} \sum_{k=1}^{K} \sum_{m=1}^{M} \frac{|B_{m,k}|}{n_k} \left| \text{acc}(B_{m,k}) - \text{conf}(B_{m,k}) \right| \tag{13}$$

where $K$ is the number of classes, $B_{m,k}$ is the $m$-th confidence bin for class $k$, and $n_k$ is the number of samples in class $k$. We also use $M = 15$ bins for CECE.

**Adaptive Calibration Error (ACE):** In addition to fixed-bin calibration metrics, we also report Adaptive Calibration Error (ACE) (Nixon et al., 2019). ACE first calculates the calibration error independently for each category and then averages the results. Within each category, the 0–1 confidence intervals are not uniformly partitioned. Instead, the predicted probabilities for the samples in that category are sorted and divided into equal-sized bins. This approach prevents estimation instability caused by insufficient samples in certain confidence intervals. Formally, ACE is defined as:

$$\text{ACE} = \frac{1}{K} \sum_{k=1}^{K} \sum_{m=1}^{M} \frac{1}{M} \left| \text{acc}(B_{m,k}) - \text{conf}(B_{m,k}) \right| \tag{14}$$

where $K$ is the number of classes, $B_{m,k}$ denotes the $m$-th adaptive confidence bin for class $k$, constructed by sorting predictions and equally partitioning them. In our experiments, we use $M = 15$ adaptive bins for ACE.

## C.2. Datasets and Preprocessing

### C.2.1. SYNTHETIC DISTRIBUTION SHIFT DATASETS

**CIFAR-10/100** (Krizhevsky & Hinton, 2009): Both datasets consist of 50,000 training and 10,000 test images of size $32 \times 32$. We create a validation set by randomly sampling 10% of the training data. For training, we apply standard data augmentation: `RandomCrop(32, padding=4)` and `RandomHorizontalFlip`. All images are normalized with mean $(0.4914, 0.4822, 0.4465)$ and standard deviation $(0.2023, 0.1994, 0.2010)$.

**Tiny-ImageNet** (Le & Yang, 2015): This dataset contains 100,000 training and 10,000 test images from 200 classes, downsized to $64 \times 64$. We form a validation set by sampling 50 images per class from the training set. Preprocessing is identical to CIFAR, but with a crop size of 64.

**Corrupted Test Sets**: For evaluating robustness to synthetic shifts, we use CIFAR-10/100-C and Tiny-ImageNet-C (Hendrycks & Dietterich, 2019). These test sets apply 15 common corruption types ( *brightness, contrast, defocus blur,elastic transform, fog, frost, gaussian noise, glass blur, impulse noise, JPEG compression, motion blur,pixelate, shot noise, snow,and zoom blur.* at 5 severity levels, as illustrated in Figure 9. Our reported results are averaged over all 15 corruption types and 5 severity levels.

### C.2.2. REAL-WORLD DISTRIBUTION SHIFT DATASETS (WILDS)

We use three datasets from the WILDS benchmark (Koh et al., 2021), following their official data splits and evaluation protocols.

**iWildCam:** A multi-class classification dataset with 182 animal species and 323,847 camera trap images (204,888 training,

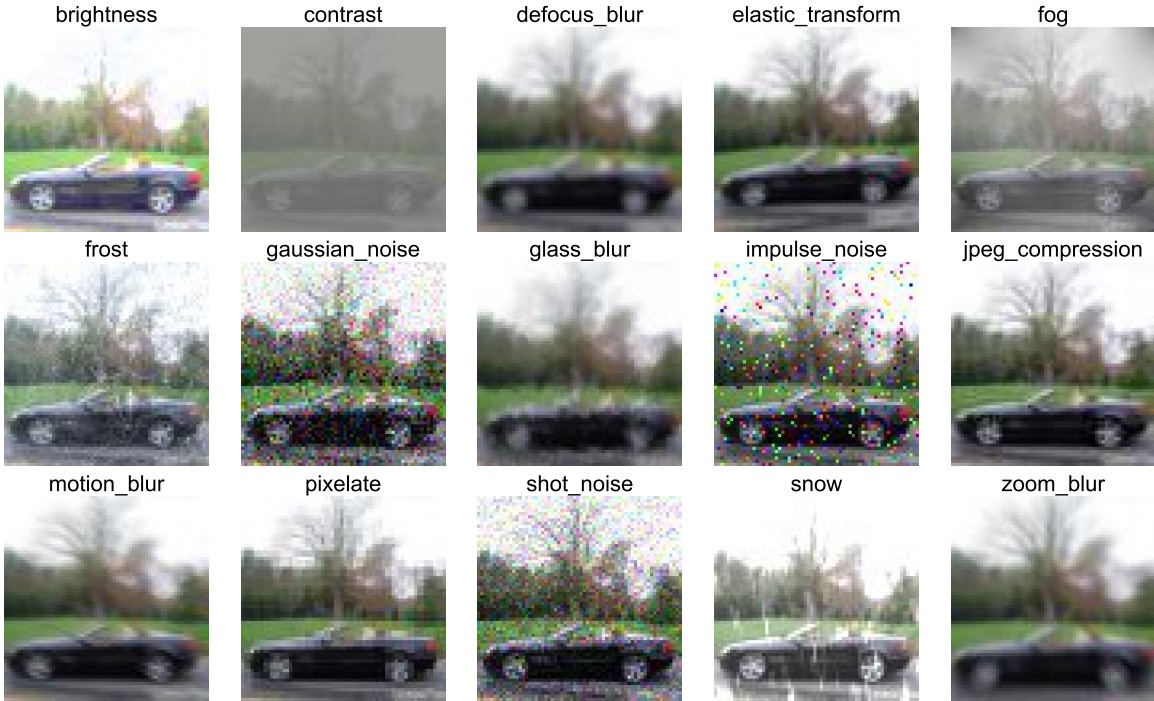

*Figure 9.* Examples of different corruption applied to Tiny-ImageNet images at severity level 3.

23,014 validation, 95,945 test) at resolution $448 \times 448$. The distribution shift occurs across different camera deployment locations, where cameras differ in angles, lighting conditions, backgrounds, and vegetation.

**FMoW (Functional Map of the World):** A multi-class classification dataset with 62 land use categories and 470,386 satellite images (362,538 training, 52,186 validation, 55,662 test) at resolution $224 \times 224$. It exhibits mixed distribution shifts where images vary by capture time and geographic regions, leading to temporal and geographical distribution differences.

**Camelyon17:** A binary classification dataset for tumor tissue identification with 449,874 image patches (302,436 training, 69,137 validation, 78,301 test) at resolution $96 \times 96$. The domain shift occurs across different hospitals, where data collection and processing methods vary.

For all WILDS datasets, we apply minimal preprocessing consisting of image resizing to the specified resolution, tensor conversion, and normalization with ImageNet statistics (mean $(0.485, 0.456, 0.406)$, standard deviation $(0.229, 0.224, 0.225)$). No data augmentation is applied during either training or testing phases.

### C.2.3. MULTI-DOMAIN SEMANTIC SHIFT DATASET

**Office-Home** (Venkateswara et al., 2017): A multi-domain classification dataset with 65 object categories commonly found in office and home environments. The dataset contains 15,500 images distributed across four distinct domains: *Art* (artistic depictions), *Clipart* (clipart images), *Product* (images without background), and *Real-World* (images captured with regular cameras). The domain shift arises from different visual styles and image capture methods across these four domains. Each domain contains approximately 3,800-4,300 images. We follow the standard protocol where each domain is split into training and test sets. Images are resized to $224 \times 224$ resolution. We apply standard preprocessing including tensor conversion and normalization with ImageNet statistics (mean $(0.485, 0.456, 0.406)$, standard deviation $(0.229, 0.224, 0.225)$). The evaluation focuses on the model's ability to maintain calibration across different visual domains representing semantic shifts in object appearance.

### C.3. Model Architectures and Training Configurations

For **CIFAR-10/100 and Tiny-ImageNet**, we train models from scratch. The architectures include ResNet-50/110, DenseNet-121, and Wide-ResNet-26, following the setup in Mukhoti et al. (2020). For **WILDS datasets**, we fine-tune models

pre-trained on ImageNet that are available in *torchvision* (Torch Contributors, 2017), following the standard practice for this benchmark. For **Office-Home datasets**, we also fine-tune ResNet-50 models pre-trained on ImageNet.

The architectures and training hyperparameters are summarized in Table 6 and Table 7.

*Table 6.* Training configurations for CIFAR-10/100, Tiny-ImageNet and Office-Home.

| Parameter | CIFAR-10/100 | Tiny-ImageNet | Office-Home |
|---|---|---|---|
| Optimizer | SGD | SGD | SGD |
| Learning Rate | 0.1 | 0.1 | 0.1 |
| Momentum | 0.9 | 0.9 | 0.9 |
| Weight Decay | 5e-4 | 5e-4 | 5e-4 |
| Batch Size | 128 | 128 | 256 |
| Total Epochs | 350 | 100 | 90 |
| LR Decay at | 150, 250 | 40, 60 | 50, 70 |
| Pre-trained | No | No | Yes |

*Table 7.* Training configurations for WILDS datasets.

| Parameter | iWildCam | FMoW | Camelyon17 |
|---|---|---|---|
| Model | ResNet-50 | DenseNet-121 | DenseNet-121 |
| Pre-trained | ImageNet-1K | ImageNet-1K | ImageNet-1K |
| Optimizer | Adam | Adam | SGD |
| Learning Rate | 3e-5 | 1e-4 | 1e-3 |
| Weight Decay | 0.0 | 0.0 | 1e-2 |
| Momentum | - | - | 0.9 |
| Batch Size | 32 | 64 | 512 |
| Epochs | 10 | 60 | 12 |
| Scheduler | None | StepLR (step=1, $\gamma$=0.96) | None |

## C.4. Implementation Details of Methods

### C.4.1. BASELINE METHODS

We compare against a comprehensive set of training-time calibration methods. Implementation details and sources are provided in Table 8. For post-hoc calibration, we use Temperature Scaling (TS), where the temperature $T$ is optimized on the validation set via grid search over $\{0.1, 0.11, \ldots, 5.0\}$.

*Table 8.* Baseline methods and their implementation details.

| Method | Key Hyperparameters | Implementation Source |
|---|---|---|
| Cross Entropy (CE) | Standard loss function | PyTorch standard |
| Label Smoothing (LS) | Smoothing parameter $\alpha = 0.05$ | Official MaxEnt implementation[1] |
| Mixup | Beta distribution parameter $\alpha = 0.3$ | Official implementation[2] |
| AugMix | Default parameters from official repo | Official implementation[3] |
| FLSD-53 | Focal loss with official settings | Official implementation[4] |
| Dual Focal Loss (DFL) | Official $\gamma$ values per dataset | Official implementation[5] |
| MaxEnt | Regularization strength $\gamma = 1$ | Official implementation[1] |
| BSCE-GRA | Manually implemented based on the paper | - |

---

[1] https://github.com/dexterdley/MaxEnt-Loss
[2] https://github.com/facebookresearch/mixup-cifar10
[3] https://github.com/google-research/augmix
[4] https://github.com/torrvision/focal_calibration
[5] https://github.com/Linwei94/ICML2023-DualFocalLoss

C.4.2. OUR METHOD (FGR)

Our Frequency-aware Gradient Rectification (FGR) method is built upon a dual-objective framework, using Soft-Binned ECE (Soft-ECE) as the calibration loss $\mathcal{L}_{\text{calib}}$.

**Calibration Loss: Soft-ECE.** Standard ECE is non-differentiable due to its hard binning process. We employ Soft-ECE (Karandikar et al., 2021), which replaces hard assignment with a differentiable soft binning mechanism. Given $M$ bins with centers $\xi_m = (m - 0.5)/M$, the soft membership score $u_{im}$ for a prediction with confidence $\hat{p}_i$ to bin $m$ is defined using a softmax function controlled by a temperature parameter $t > 0$:

$$u_{im} = \text{softmax}_m \left( -\frac{(\hat{p}_i - \xi_m)^2}{t} \right). \tag{15}$$

Based on these soft memberships, the statistics for each bin $S_m$ are redefined as follows:

- **Soft bin count:** $|S_m| = \sum_i u_{im} + \epsilon$, where $\epsilon$ is a small constant (e.g., $10^{-12}$) for numerical stability.

- **Soft bin accuracy:** $\text{acc}(S_m) = \frac{1}{|S_m|} \sum_i u_{im} \cdot \mathbb{1}[\hat{y}_i = y_i]$.

- **Soft bin confidence:** $\text{conf}(S_m) = \frac{1}{|S_m|} \sum_i u_{im} \cdot \hat{p}_i$.

The final Soft-ECE loss is then calculated using these differentiable statistics. Specifically, we use the $L_2$ variant of Soft-Binned ECE:

$$\mathcal{L}_{\text{calib}} = \left( \sum_{m=1}^{M} \frac{|S_m|}{N} |\text{acc}(S_m) - \text{conf}(S_m)|^2 \right)^{1/2}, \tag{16}$$

where $N$ is the total number of samples in the batch. In our implementation, we use a fixed temperature of $t = 0.1$ for all experiments.

**Hyperparameters.** Our FGR method has three main tunable hyperparameters:

- **Main Loss Focusing Parameter ($\gamma$):** We use Dual Focal Loss (DFL) as $\mathcal{L}_{\text{main}}$. The focusing parameter $\gamma$ is tuned per dataset and architecture. For CIFAR and Tiny-ImageNet, the values depend on the training strategy (see Tables 9 and 10). For WILDS datasets, we use $\gamma = 7$ for iWildCam, $\gamma = 5$ for FMoW, and $\gamma = 5$ for Camelyon17 in the training-from-scratch setting.

- **Filtering Ratio ($\rho$):** For both synthetic and real offset datasets, this ratio is fixed at $\rho = 0.05$ for all experiments, meaning 5% of the training data is filtered in each epoch. For the semantic offset dataset Office-Home, $\rho$ is set to 0.1.

- **DCT Compression Parameter ($\lambda$):** To introduce diversity, $\lambda$ is randomly sampled from $\{15, 18, 25\}$ for each filtered image.

**Training Strategies.** To ensure a fair comparison with other methods, the results reported in the main body of the paper for all datasets (CIFAR-10/100, Tiny-ImageNet, and WILDS) are based on **training from scratch**. This involves training the model for its full duration (e.g., 350 epochs for CIFAR), using only the main loss for the first 200 epochs and then introducing our full FGR mechanism from epoch 201 onwards.

Additionally, to demonstrate the practicality and computational efficiency of our method, we conducted experiments using a **two-stage fine-tuning** strategy on CIFAR-10/100 and Tiny-ImageNet. In this setup, a model is first fully trained with standard Cross-Entropy. Then, its backbone is frozen, and only the classification head is fine-tuned for a small number of epochs using our FGR framework. This approach achieves comparable or even superior performance with significantly less training time.

The optimal $\gamma$ values differ between these two strategies, as detailed in the tables 9, 10 below.

*Table 9.* Focusing parameter $\gamma$ for FGR when **training from scratch**.

| Architecture | CIFAR-10 | CIFAR-100 | Tiny-ImageNet |
|---|---|---|---|
| ResNet-50 | 3.0 | 2.0 | 3.0 |
| ResNet-110 | 4.5 | 3.0 | - |
| DenseNet-121 | 4.0 | 1.5 | 2.0 |
| WideResNet-26 | 1.5 | 1.2 | - |

*Table 10.* Focusing parameter $\gamma$ for FGR in the **two-stage fine-tuning** strategy.

| Architecture | CIFAR-10 | CIFAR-100 | Tiny-ImageNet |
|---|---|---|---|
| ResNet-50 | 5.0 | 3.5 | 3.6 |
| ResNet-110 | 1.5 | 4.0 | - |
| DenseNet-121 | 5.5 | 3.4 | - |
| WideResNet-26 | 1.5 | 2.1 | - |

# D. Additional Experimental Results

## D.1. Hyperparameter Sensitivity Analysis

We conduct a sensitivity analysis for the three main hyperparameters of our FGR method: the main loss focusing parameter $\gamma$, the filtering ratio $\rho$, and the DCT compression parameter $\lambda$. All experiments are performed on CIFAR-10/100 using the ResNet-50 architecture.

**Focusing Parameter ($\gamma$).** We analyze the impact of the focusing parameter $\gamma$ from the main loss (Dual Focal Loss). As shown in Figure 10, we vary $\gamma$ from 1 to 7 while keeping the filtering ratio fixed at $\rho = 0.05$. The results indicate that an appropriate $\gamma$ value is crucial for balancing accuracy and calibration. For each dataset and architecture combination, we select the $\gamma$ that yields the best ECE on the in-distribution validation set, and use this value for all reported experiments.

**Filtering Ratio ($\rho$).** Figure 11 (left) shows the effect of the filtering ratio $\rho$. We observe that the best calibration performance is achieved when $\rho$ is in the range of $[0.05, 0.1]$. A small ratio ensures that enough original samples are available for the calibration objective, while still providing sufficient frequency-filtered samples for the main objective to learn robust features. An excessively large $\rho$ can harm in-distribution performance by reducing the number of clean samples used for calibration. We fix $\rho = 0.05$ for all our experiments.

**DCT Compression Parameter ($\lambda$).** The sensitivity to the DCT compression parameter $\lambda$ is shown in Figure 11 (right). Smaller values of $\lambda$ correspond to more aggressive filtering of high-frequency components. The results show that while aggressive filtering can improve calibration under distribution shift (OOD ECE), it may slightly degrade in-distribution (ID) calibration. The optimal trade-off is found when $\lambda$ is in the range of $[15, 25]$. To introduce diversity and avoid overfitting to a single filtering level, we randomly sample $\lambda$ from $\{15, 18, 25\}$ for each filtered image in our experiments.

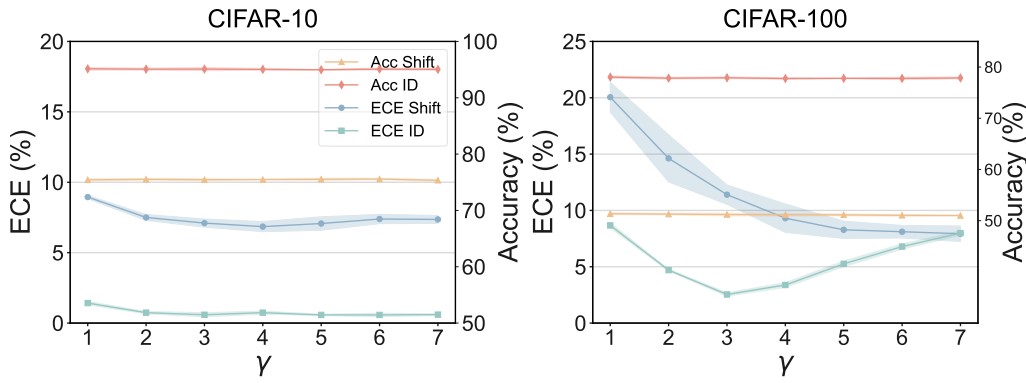

*Figure 10.* Hyperparameter sensitivity analysis for the focusing parameter $\gamma$ on CIFAR-10 and CIFAR-100 with ResNet-50.

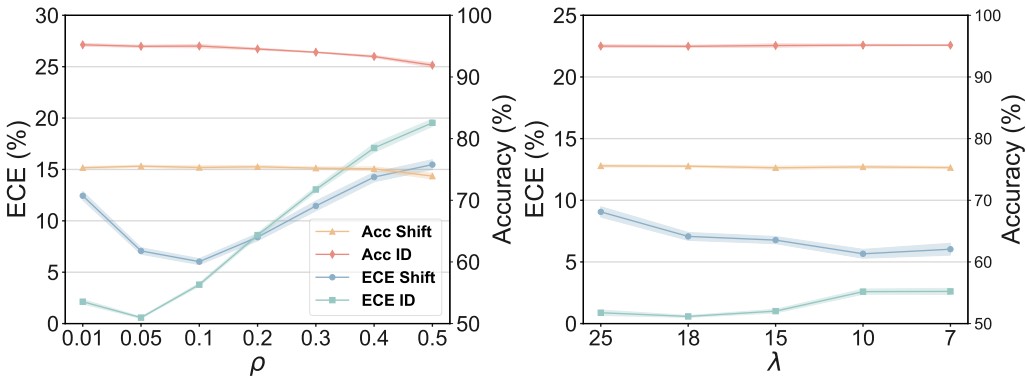

*Figure 11.* Sensitivity analysis of filtering ratio $\rho$ (left) and DCT compression parameter $\lambda$ (right) using ResNet-50 on CIFAR-10 and CIFAR-10-C.

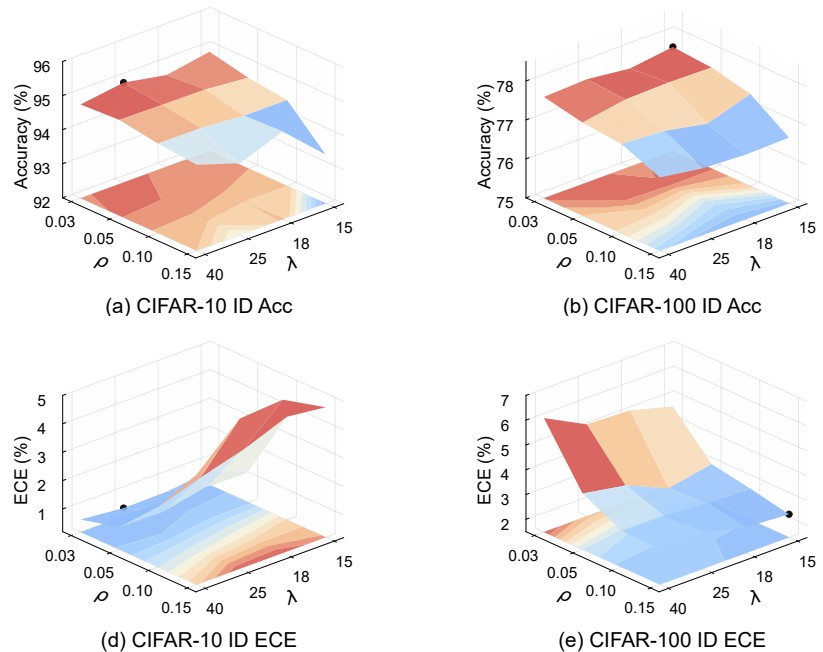

*Figure 12.* Joint sensitivity of FGR to filtering ratio $\rho$ and DCT compression parameter $\lambda$ on in-distribution evaluation. For CIFAR datasets, we report clean test accuracy and ECE; for iWildCam, we use the validation split as the in-distribution-side model-selection proxy.

We further conduct a two-dimensional sensitivity analysis over the filtering ratio $\rho$ and the DCT compression parameter $\lambda$. We evaluate $\rho \in \{0.03, 0.05, 0.10\}$ and $\lambda \in \{15, 18, 25\}$ on CIFAR-10, CIFAR-100, and iWildCam. All results are averaged over three random seeds. For CIFAR-10 and CIFAR-100, we report both clean-set performance and corrupted-set performance on CIFAR-10-C and CIFAR-100-C. For iWildCam, we report performance on the shifted test split.

Figures 12 and 13 show a consistent pattern across datasets. The filtering ratio $\rho$ is the dominant factor controlling the ID–OOD trade-off. Smaller $\rho$ better preserves clean-side performance, whereas larger $\rho$ generally improves shifted calibration but may introduce a larger clean-side cost. By comparison, the effect of $\lambda$ is weaker within the moderate range $\{15, 18, 25\}$, suggesting that FGR is not brittle to the exact compression strength. Overall, $\rho = 0.05$ provides a strong default trade-off on CIFAR-10 and iWildCam, while CIFAR-100 benefits slightly more from stronger filtering. These results support using $\rho$ as the primary practical knob for controlling the robustness-calibration trade-off.

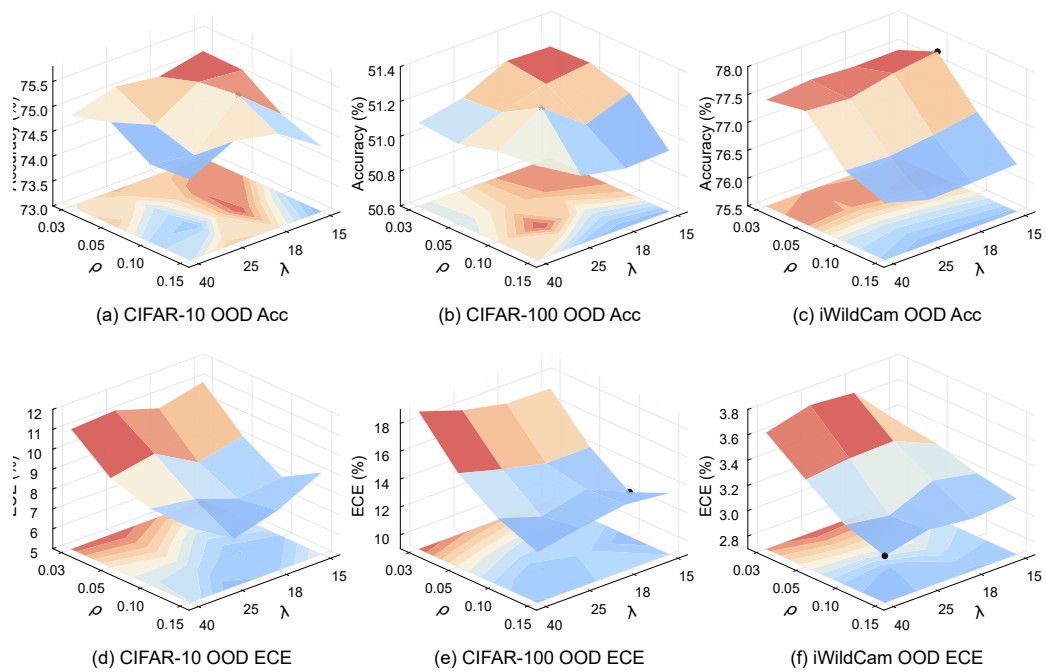

*Figure 13.* Joint sensitivity of FGR to filtering ratio $\rho$ and DCT compression parameter $\lambda$ under distribution shift. Results are shown for CIFAR-10-C, CIFAR-100-C, and iWildCam. Across datasets, $\rho$ acts as the main accuracy-calibration trade-off parameter, while $\lambda$ has a comparatively smaller effect within the tested range.

## D.2. Comparison of Training Strategies

To ensure a fair comparison with baseline methods, all results reported in the main body of the paper are based on a **training from scratch** strategy. This involves training the model for its full duration (e.g., 350 epochs for CIFAR), introducing our FGR mechanism after an initial warm-up phase (e.g., from epoch 201).

In this section, we present an alternative **two-stage fine-tuning** strategy, designed to demonstrate the practicality and computational efficiency of our method. In this setup, a model is first fully trained with standard Cross-Entropy. Then, its backbone is frozen, and only the classification head is fine-tuned for a small number of epochs using our FGR framework.

First, Tables 11 and 12 provide a direct comparison between the two strategies on CIFAR-10/100 using a ResNet-50 architecture. The results show that both approaches achieve comparable performance on both in-distribution and distribution shift test sets.

*Table 11.* Comparison of training strategies on in-distribution test sets (ResNet-50).

| Strategy | CIFAR-10 | | CIFAR-100 | |
|---|---|---|---|---|
| | ACC | ECE | ACC | ECE |
| Training from Scratch | 94.97 | 0.65 | 78.30 | 2.84 |
| Two-stage Fine-tuning | 95.06 | **0.58** | 78.01 | **2.49** |

*Table 12.* Comparison of training strategies on distribution shift test sets (ResNet-50).

| Strategy | CIFAR-10-C | | CIFAR-100-C | |
|---|---|---|---|---|
| | ACC | ECE | ACC | ECE |
| Training from Scratch | 75.23 | **6.78** | 50.89 | 10.50 |
| Two-stage Fine-tuning | 75.53 | 7.07 | 51.33 | **9.94** |

Furthermore, to demonstrate that this efficient two-stage strategy is competitive against state-of-the-art methods, Table 13 presents a detailed comparison on synthetic distribution shift benchmarks using the ResNet-50 model trained with our two-stage approach. The results show that our method consistently outperforms all baselines in calibration metrics (ECE and CECE) under distribution shift, confirming the effectiveness of the fine-tuning strategy.

*Table 13.* Test scores (%) of different methods on synthetic (top) and real-world (bottom) distribution shift test sets. For synthetic datasets, results are averaged over 15 corruption types across 5 severity levels. The "w/ TS" columns show ECE and CECE values with temperature scaling post-hoc calibration. The best average scores are highlighted in **bold**.

| Loss Fn. | **CIFAR-10-C** / ResNet-50 | | | | | **CIFAR-100-C** / ResNet-50 | | | | | **Tiny ImageNet-C** / ResNet-50 | | | | |
|---|---|---|---|---|---|---|---|---|---|---|---|---|---|---|---|
| | ACC. | ECE | w/ TS | CECE | w/ TS | ACC. | ECE | w/ TS | CECE | w/ TS | ACC. | ECE | w/ TS | CECE | w/ TS |
| CE | 74.59 | 22.60 | 15.20 | 4.71 | 3.56 | 51.03 | 38.29 | 14.11 | 0.85 | 0.47 | 24.29 | 35.52 | 14.25 | 0.52 | 0.39 |
| LS-0.05 | 75.15 | 13.68 | 15.19 | 3.33 | 3.52 | 50.73 | 11.05 | 10.72 | 0.46 | **0.45** | 24.65 | **13.95** | 15.06 | **0.37** | 0.40 |
| FLSD-53 | 73.45 | 14.65 | 13.86 | 3.74 | 3.66 | 49.31 | 20.63 | 13.27 | 0.61 | 0.55 | 21.90 | 17.13 | 26.40 | 0.43 | 0.48 |
| DFL | 72.61 | 14.02 | 13.84 | 3.73 | 3.71 | 49.80 | 12.15 | 12.61 | 0.52 | 0.52 | 23.67 | 20.31 | 16.71 | 0.44 | 0.42 |
| MaxEnt M | 75.20 | 11.21 | 11.38 | 3.20 | 3.22 | 47.71 | 16.73 | 15.16 | 0.62 | 0.61 | 19.61 | 27.24 | 17.48 | 0.50 | 0.45 |
| BSCE-GRA | 72.23 | 13.29 | 13.47 | 3.68 | 3.70 | 48.46 | 16.42 | 13.61 | 0.58 | 0.56 | 22.95 | 20.42 | 47.77 | 0.45 | 0.63 |
| **FGR** | 75.53 | **7.07** | **7.07** | **2.70** | **2.70** | 51.33 | **9.94** | 10.45 | **0.44** | 0.47 | 23.57 | 16.46 | **12.96** | 0.41 | **0.38** |

While both training strategies demonstrate similar effectiveness, we recommend the two-stage fine-tuning approach for practical implementation. This preference is motivated by two key advantages: (1) **Strong Foundation**: The initial cross-entropy pre-training establishes robust feature representations, providing a solid base for subsequent calibration improvements. (2) **Training Efficiency**: The fine-tuning approach requires significantly less computational time, as only the lightweight classification head is retrained with our FGR objective. This makes it a more practical and scalable choice for real-world applications without compromising performance.

### D.3. In-Distribution Calibration Performance

To ensure that our method does not harm calibration on clean in-distribution data, we evaluate all methods on the original test sets of CIFAR-10/100 and Tiny-ImageNet. As shown in Table 14, our method maintains competitive or superior ECE performance in most cases, even in the absence of distribution shifts, demonstrating its general applicability.

*Table 14.* ECE↓ (%) before and after temperature scaling for different methods on the in-distribution test set. In the experiment, ECE was evaluated for different methods before (Pre T) and after (Post T) temperature scaling.

| Dataset | Model | CE | | FLSD-53 | | DFL | | MaxEnt | | BSCE-GRA | | FGR | |
|---|---|---|---|---|---|---|---|---|---|---|---|---|---|
| | | Pre T | Post T | Pre T | Post T | Pre T | Post T | Pre T | Post T | Pre T | Post T | Pre T | Post T |
| CIFAR-10 | ResNet-50 | 4.16 | 1.14 | 1.28 | 1.01 | 1.37 | 1.28 | 1.42 | 1.49 | **0.60** | 0.70 | 0.65 | 0.68 |
| | ResNet-110 | 4.46 | 1.17 | 1.26 | 0.94 | 1.49 | 0.92 | 1.28 | 1.28 | 1.16 | 1.07 | 0.90 | **0.84** |
| | DenseNet-121 | 4.62 | 1.75 | 1.19 | 0.98 | 1.18 | 0.65 | 1.07 | 1.18 | 1.56 | 1.05 | 0.97 | **0.64** |
| | WideResnet-26 | 3.49 | 1.26 | 2.16 | 1.19 | 1.87 | 1.11 | 1.27 | 1.05 | 2.48 | 1.16 | 1.23 | **0.58** |
| CIFAR-100 | ResNet-50 | 17.11 | 2.31 | 4.71 | 2.41 | 1.48 | 1.48 | 4.56 | 3.51 | 2.91 | **0.99** | 2.84 | 2.49 |
| | ResNet-110 | 18.28 | 4.27 | 6.75 | 3.99 | 3.27 | 3.27 | 3.72 | 3.94 | 3.85 | **2.96** | 3.48 | 3.57 |
| | DenseNet-121 | 18.95 | 3.57 | 2.93 | **1.42** | 1.96 | 1.51 | 1.71 | 1.75 | 1.68 | 1.54 | 3.32 | 3.34 |
| | WideResnet-26 | 14.75 | 2.86 | 2.06 | 2.41 | 2.77 | **1.64** | 2.38 | 2.48 | 2.00 | 1.86 | 3.01 | 2.93 |
| Tiny-ImageNet | ResNet-50 | 13.60 | 2.91 | 1.92 | 8.67 | 2.82 | 1.57 | 7.75 | 1.82 | 2.05 | 15.15 | 2.26 | **1.02** |
| | DenseNet-121 | 8.26 | 15.01 | 3.81 | 15.17 | 3.68 | 1.37 | 4.12 | 1.32 | 6.01 | 1.62 | 3.42 | **1.22** |

### D.4. Effect of Alternative Calibration Losses in FGR

We further evaluate the flexibility of the FGR framework by replacing Soft-ECE with other differentiable calibration losses, including Soft-AvUC, MMCE, and Brier Score. Experiments are conducted on CIFAR-10/100 under both in-distribution (ID) and corrupted (OOD) settings using ResNet-50.

The results in Tables 15 and 16 demonstrate the versatility of our gradient rectification framework. Across different calibration losses, FGR consistently outperforms the CE baseline in both ID and OOD scenarios. Notably:

- **Soft-ECE** achieves the best overall balance, with strong performance on both ID and OOD ECE metrics.

- **Soft-AvUC** shows competitive OOD performance, particularly on CIFAR-10-C (ECE: 6.47%).

- **MMCE** and **Brier-Score** also yield improvements, confirming that our projection-based gradient rectification is loss-agnostic.

These results validate that FGR provides a general framework for calibration improvement, not limited to any specific calibration objective. The choice of calibration loss can be tailored to specific application requirements without compromising the core benefits of our approach.

*Table 15.* In-distribution results on CIFAR-10 and CIFAR-100 with ResNet-50 using different calibration losses within the FGR framework. All metrics are reported in percentage (%).

| | CIFAR-10 | | | | | CIFAR-100 | | | | |
|---|---|---|---|---|---|---|---|---|---|---|
| Method | ACC ↑ | ECE ↓ | TS-ECE ↓ | ACE ↓ | TS-ACE ↓ | ACC ↑ | ECE ↓ | TS-ECE ↓ | ACE ↓ | TS-ACE ↓ |
| Cross-Entropy | 95.27 | 4.17 | 1.14 | 0.696 | 0.580 | 78.04 | 17.11 | 2.40 | 0.190 | 0.184 |
| FGR (Soft-ECE) | 95.11 | **0.86** | 0.66 | 0.534 | 0.485 | 77.79 | **2.53** | **2.27** | 0.182 | 0.170 |
| FGR (Soft-AvUC) | 95.06 | 6.07 | 1.21 | 1.299 | 0.513 | 77.84 | 3.28 | 2.49 | 0.198 | **0.168** |
| FGR (MMCE) | 95.16 | 1.06 | 0.79 | **0.523** | 0.583 | 77.87 | 2.62 | 2.76 | 0.183 | 0.189 |
| FGR (Brier-Score) | 95.14 | 2.77 | **0.52** | 0.794 | **0.485** | 77.90 | 3.02 | 2.63 | 0.194 | 0.176 |

*Table 16.* Out-of-distribution results on CIFAR-10-C and CIFAR-100-C with ResNet-50 using different calibration losses within the FGR framework. All metrics are reported in percentage (%).

| | CIFAR-10-C | | | | | CIFAR-100-C | | | | |
|---|---|---|---|---|---|---|---|---|---|---|
| Method | ACC ↑ | ECE ↓ | TS-ECE ↓ | ACE ↓ | TS-ACE ↓ | ACC ↑ | ECE ↓ | TS-ECE ↓ | ACE ↓ | TS-ACE ↓ |
| Cross-Entropy | 74.59 | 22.60 | 15.20 | 4.228 | 3.448 | 51.03 | 38.29 | 14.11 | 0.679 | 0.481 |
| FGR (Soft-ECE) | 75.18 | 6.88 | 7.57 | 2.717 | 2.735 | 51.33 | 9.93 | 10.75 | **0.448** | **0.448** |
| FGR (Soft-AvUC) | 75.13 | **6.47** | 7.22 | 2.971 | **2.684** | 51.26 | **9.54** | 11.69 | 0.457 | 0.458 |
| FGR (MMCE) | 75.27 | 8.12 | **7.11** | 2.756 | 2.727 | 51.30 | 10.96 | **10.54** | 0.456 | 0.456 |
| FGR (Brier-Score) | 75.13 | 5.89 | 7.93 | **2.728** | 2.712 | 51.15 | 9.83 | 11.07 | 0.460 | 0.460 |

### D.5. Sensitivity to Batch Size

We evaluate the sensitivity of FGR to different batch sizes to verify that its calibration improvements are not contingent on a specific batch configuration. Across a wide range of batch sizes, FGR exhibits consistent calibration performance, indicating that the proposed gradient rectification mechanism is robust to this implementation choice.

We conduct batch-size sensitivity experiments on CIFAR-10/100 and their corresponding corrupted datasets. Our baseline batch size is 128. Since directly reducing the batch size leads to a significant drop in accuracy, we employ $4\times/2\times$ gradient accumulation for batch sizes 32/64 to maintain equivalent optimization strides. Experimental results are presented in the table below.

Since FGR's gradient rectification is based on conflict detection across the entire batch, smaller batch sizes (e.g., 32 and 64) can capture more fine-grained conflicts between the main loss and ID calibration loss. This yields strong OOD calibration performance while maintaining in-distribution performance. Larger batch sizes (e.g., 512) may hide certain local conflicts, leading to a clear performance drop. Although smaller batches perform better, training time also increases (32 vs. 128: 4.83h vs. 1.7h). To achieve a better balance between performance and cost, we recommend using a medium-sized batch size of 128.

### D.6. Computational Complexity Analysis

We compare the computational overhead of our Frequency-aware Gradient Rectification (FGR) against representative training-time calibration baselines (CE, LS, FLSD-53, DFL, MaxEnt, BSCE-GRA). We report the empirical wall-clock

*Table 17.* Sensitivity to batch size on in-distribution test sets. Calibration metrics are multiplied by 100.

| | CIFAR-10 | | | | | CIFAR-100 | | | | |
|---|---|---|---|---|---|---|---|---|---|---|
| **Batch Size** | ACC↑ | ECE↓ | TS-ECE↓ | ACE↓ | TS-ACE↓ | ACC↑ | ECE↓ | TS-ECE↓ | ACE↓ | TS-ACE↓ |
| 32 | 94.26 | 1.37 | 1.14 | 0.388 | 0.306 | 78.14 | 1.42 | 1.47 | 0.130 | 0.138 |
| 64 | 94.23 | **0.80** | 0.84 | 0.301 | 0.306 | 77.92 | 1.74 | 2.13 | 0.134 | 0.143 |
| 128 | 94.09 | 0.82 | **0.72** | 0.328 | 0.326 | 78.35 | 1.97 | 1.79 | 0.185 | 0.181 |
| 256 | 94.30 | 1.35 | 0.72 | **0.231** | **0.217** | 77.75 | 1.76 | 1.53 | **0.093** | **0.115** |
| 512 | 94.10 | 2.25 | 1.28 | 0.268 | 0.237 | 76.24 | 3.18 | 2.10 | 0.105 | 0.123 |

*Table 18.* Sensitivity to batch size under distribution shift (CIFAR-C). Calibration metrics are multiplied by 100.

| | CIFAR-10-C | | | | | CIFAR-100-C | | | | |
|---|---|---|---|---|---|---|---|---|---|---|
| **Batch Size** | ACC↑ | ECE↓ | TS-ECE↓ | ACE↓ | TS-ACE↓ | ACC↑ | ECE↓ | TS-ECE↓ | ACE↓ | TS-ACE↓ |
| 32 | **77.01** | **7.76** | 10.43 | **2.80** | **2.91** | **52.58** | 10.55 | **9.71** | **0.48** | **0.48** |
| 64 | 74.21 | 9.99 | **9.80** | 3.27 | 3.26 | 51.87 | **10.46** | 9.66 | 0.48 | 0.70 |
| 128 | 73.57 | 12.20 | 11.63 | 3.56 | 3.53 | 51.02 | 10.15 | 10.56 | 0.50 | 0.50 |
| 256 | 73.14 | 13.27 | 11.81 | 3.60 | 3.51 | 51.37 | 11.75 | 10.50 | 0.48 | 0.48 |
| 512 | 72.52 | 14.81 | 12.33 | 3.74 | 3.56 | 48.63 | 14.41 | 10.36 | 0.52 | 0.51 |

measurements. All experiments are conducted under identical hardware and software settings: **GPU:** NVIDIA GeForce RTX 4090 24GB; **Software:** PyTorch v2.2, CUDA 12.1; **Precision:** AMP (Automatic Mixed Precision).

For each method we record: (a) average epoch time and (b) total training time. All measurements are performed on the CIFAR-100 dataset using a ResNet-50 architecture, following the training schedule defined in Table 6. For our two-stage FGR-FT method, we report the time for Stage 1 (CE pre-training) and Stage 2 (head fine-tuning) separately.

*Table 19.* Empirical training time comparison on CIFAR-100 with ResNet-50. We report the average time per epoch and the total training time. For our two-stage FGR-FT, the reported time corresponds only to the second stage (head fine-tuning), assuming a pre-trained CE model is available.

| Metric | CE | LS-0.05 | Mixup | AugMix | DFL | MaxEnt | BSCE-GRA | FGR-Scratch | FGR-FT |
|---|---|---|---|---|---|---|---|---|---|
| $t_{epoch}$ (s) | 10 | 12 | 13 | 23 | 10 | 12 | 10 | 15 | **4** |
| $t_{total}$ (h) | 1.62 | 1.81 | 1.74 | 3.03 | 1.62 | 1.55 | 1.55 | 1.92 | **0.29** |

The results in Table 19 highlight the computational efficiency of our proposed methods. The two-stage fine-tuning strategy (FGR-FT) is exceptionally efficient, adding only a small incremental cost (0.29h) over a standard pre-trained CE model, as it only fine-tunes the lightweight classification head. The training-from-scratch strategy (FGR-Scratch) shows a modest overhead of approximately 18.5% compared to the CE baseline (1.92h vs. 1.62h). This slight increase is primarily due to the Soft-ECE loss calculation, the occasional gradient projection, and the DCT operation on a small subset of the data. Compared to other methods that introduce complex augmentations or per-sample loss modulations (e.g., AugMix), our FGR approach remains computationally competitive while delivering superior calibration performance under distribution shifts.

### D.7. Feature-Space Wasserstein Analysis

To better understand how frequency filtering changes the induced training distribution, we measure the feature-space discrepancy between the original distribution $\mathcal{D}_{orig}$ and the mixed distribution $\mathcal{D}_{mix}$. Specifically, we extract penultimate-layer features using a ResNet-50 model trained with cross-entropy on CIFAR-10, and compute the Gaussian 2-Wasserstein distance between features from $\mathcal{D}_{orig}$ and $\mathcal{D}_{mix}$ using 4096 training samples.

Tables 20 and 21 show that the distributional discrepancy is controlled primarily by the filtering ratio $\rho$. Increasing $\rho$ leads to a clear and nearly monotonic increase in the feature-space distance, indicating that filtering a larger fraction of samples induces a stronger shift between $\mathcal{D}_{orig}$ and $\mathcal{D}_{mix}$. In contrast, changing $\lambda$ within a moderate range has a much milder effect. This supports our empirical observation that $\rho$ is the dominant trade-off parameter, while $\lambda$ mainly provides a secondary

*Table 20.* Feature-space Gaussian 2-Wasserstein distance between $\mathcal{D}_{\text{orig}}$ and $\mathcal{D}_{\text{mix}}$ when varying the DCT compression parameter $\lambda$ with fixed filtering ratio $\rho = 0.1$.

| $\lambda$ | 25 | 18 | 15 | 10 | 7 |
|---|---|---|---|---|---|
| $W_2$ | 0.517 | 0.522 | 0.510 | 0.457 | 0.471 |

*Table 21.* Feature-space Gaussian 2-Wasserstein distance between $\mathcal{D}_{\text{orig}}$ and $\mathcal{D}_{\text{mix}}$ when varying the filtering ratio $\rho$ with fixed $\lambda = 15$.

| $\rho$ | 0.05 | 0.10 | 0.20 | 0.40 |
|---|---|---|---|---|
| $W_2$ | 0.300 | 0.510 | 1.018 | 1.952 |

adjustment of filtering strength.

# E. Proof of the Projection Property

## E.1. Proposition Restated

For a training step, let $\mathbf{g}_{\text{main}} = \nabla_\theta \mathcal{L}_{\text{main}}(\theta; \mathcal{D}_{\text{mix}})$ and $\mathbf{g}_{\text{calib}} = \nabla_\theta \mathcal{L}_{\text{calib}}(\theta; \mathcal{D}_{\text{orig}})$. Ignoring the small numerical stabilizer used in implementation, FGR uses

$$\mathbf{g}_{\text{FGR}} = \begin{cases} \mathbf{g}_{\text{main}}, & \text{if } \mathbf{g}_{\text{main}}^\top \mathbf{g}_{\text{calib}} \geq 0, \\ \mathbf{g}_{\text{main}} - \dfrac{\mathbf{g}_{\text{main}}^\top \mathbf{g}_{\text{calib}}}{\|\mathbf{g}_{\text{calib}}\|_2^2} \mathbf{g}_{\text{calib}}, & \text{otherwise.} \end{cases} \tag{17}$$

Define the first-order feasible set

$$\mathcal{C}_{\text{ID}} = \{\mathbf{g} \mid \mathbf{g}^\top \mathbf{g}_{\text{calib}} \geq 0\}. \tag{18}$$

Then $\mathbf{g}_{\text{FGR}}$ solves

$$\mathbf{g}_{\text{FGR}} = \arg\min_{\mathbf{g} \in \mathcal{C}_{\text{ID}}} \|\mathbf{g} - \mathbf{g}_{\text{main}}\|_2^2. \tag{19}$$

## E.2. Proof

*Proof.* If $\mathbf{g}_{\text{main}}^\top \mathbf{g}_{\text{calib}} \geq 0$, then $\mathbf{g}_{\text{main}} \in \mathcal{C}_{\text{ID}}$. Since the objective is the squared Euclidean distance to $\mathbf{g}_{\text{main}}$, the feasible point $\mathbf{g} = \mathbf{g}_{\text{main}}$ attains the minimum value zero. Therefore, $\mathbf{g}_{\text{FGR}} = \mathbf{g}_{\text{main}}$ is the minimizer.

Now consider the conflicting case $\mathbf{g}_{\text{main}}^\top \mathbf{g}_{\text{calib}} < 0$. The constrained problem is the Euclidean projection of $\mathbf{g}_{\text{main}}$ onto the closed half-space $\mathcal{C}_{\text{ID}}$. Since $\mathbf{g}_{\text{main}}$ is outside the half-space, the optimum lies on the boundary $\mathbf{g}^\top \mathbf{g}_{\text{calib}} = 0$. The Lagrangian for the equivalent constraint $-\mathbf{g}^\top \mathbf{g}_{\text{calib}} \leq 0$ is

$$\mathcal{L}(\mathbf{g}, \mu) = \|\mathbf{g} - \mathbf{g}_{\text{main}}\|_2^2 - \mu\, \mathbf{g}^\top \mathbf{g}_{\text{calib}}, \qquad \mu \geq 0. \tag{20}$$

Stationarity gives

$$2(\mathbf{g} - \mathbf{g}_{\text{main}}) - \mu \mathbf{g}_{\text{calib}} = \mathbf{0}, \tag{21}$$

and the active boundary condition gives $\mathbf{g}^\top \mathbf{g}_{\text{calib}} = 0$. Solving these two equations yields

$$\mu = -2 \frac{\mathbf{g}_{\text{main}}^\top \mathbf{g}_{\text{calib}}}{\|\mathbf{g}_{\text{calib}}\|_2^2} > 0, \tag{22}$$

and therefore

$$\mathbf{g} = \mathbf{g}_{\text{main}} - \frac{\mathbf{g}_{\text{main}}^\top \mathbf{g}_{\text{calib}}}{\|\mathbf{g}_{\text{calib}}\|_2^2} \mathbf{g}_{\text{calib}}. \tag{23}$$

This is exactly the FGR direction in the conflicting case, so the projection characterization holds in both cases.

Finally, because $\mathbf{g}_{\text{FGR}} \in \mathcal{C}_{\text{ID}}$, we have $\mathbf{g}_{\text{FGR}}^\top \mathbf{g}_{\text{calib}} \geq 0$. A first-order Taylor expansion of the calibration loss gives

$$\mathcal{L}_{\text{calib}}(\theta - \eta \mathbf{g}_{\text{FGR}}) = \mathcal{L}_{\text{calib}}(\theta) - \eta\, \mathbf{g}_{\text{calib}}^\top \mathbf{g}_{\text{FGR}} + \mathcal{O}(\eta^2) \leq \mathcal{L}_{\text{calib}}(\theta) + \mathcal{O}(\eta^2), \tag{24}$$

which proves the first-order ID calibration preservation guarantee. $\square$

## E.3. Relation to a Joint-Risk View

The projection result above is the exact statement corresponding to the implemented optimizer. A scalarized joint objective such as $\hat{\mathcal{R}}_{\mathrm{mix}} + \alpha\hat{\mathcal{R}}_{\mathrm{calib}}$ would produce an additive gradient direction $\mathbf{g}_{\mathrm{main}} + \alpha\mathbf{g}_{\mathrm{calib}}$, which is not the conditional projection used by FGR. We therefore use the joint-risk view only as high-level intuition: FGR couples the mixed-data objective with an ID calibration constraint, but it does not claim equivalence to minimizing a fixed weighted sum of losses. This distinction is important because FGR is asymmetric: the main gradient is left unchanged whenever it is already feasible, and it is modified only by the minimum Euclidean correction required to satisfy the first-order calibration constraint.

## F. Limitations and Scope

While our proposed FGR method demonstrates strong empirical performance, several limitations warrant discussion. Our study focuses exclusively on image classification under covariate shift scenarios, where the frequency-domain filtering relies on Discrete Cosine Transform (DCT). The method does not handle unknown-class out-of-distribution detection, and shows relatively limited gains on semantic shift datasets such as Office-Home compared to synthetic corruption benchmarks. FGR introduces additional training costs due to extra calibration gradient computation; however, as demonstrated in Section D.6, the overhead is controlled at approximately 18.5% for training from scratch, and our two-stage fine-tuning strategy significantly reduces this barrier by enabling efficient adaptation on pre-trained models. Promising future directions include adapting the approach to other modalities, integrating OOD detection mechanisms, and developing more effective strategies for semantic shift scenarios.

