# OpenReview forum: "Target-Agnostic Calibration under Distribution Shift with Frequency-Aware Gradient Rectification"
_ICML.cc/2026/Conference — ICML 2026 regular_

### Official Review · Reviewer_Bo8C · 2026-02-21

**Soundness:** 2
**Presentation:** 2
**Significance:** 3
**Originality:** 1
**Overall Recommendation:** 3
**Confidence:** 4

**Summary:**

The paper proposed Frequency-aware Gradient Rectification (FGR), a target-agnostic training framework to improve predictive calibration under distribution shift. It mixed original and DCT-based low-pass filtered images to reduce reliance on high-frequency shortcuts. A projection-based gradient rectification was used to enforce a first-order constraint that prevents in-distribution (ID) calibration from worsening. Experiments on synthetic, real-world, and semantic shift datasets verified the effectiveness of the method.

**Compliance With Llm Reviewing Policy:**

Affirmed.

**Final Justification:**

I thank the authors for their thoughtful rebuttal. The proposed revisions have successfully addressed my primary concerns about theoretical soundness. However, as concerns about the degree of novelty and the method's practical trade-offs still linger, I have raised my score to Weak Reject to acknowledge the substantial improvements.

**Key Questions For Authors:**

1. Could you clarify the specific novelty of your method beyond applying the gradient projection mechanism to the problem of model calibration under distribution shift?
2. There seems to be a mismatch between the algorithm analyzed in your theoretical proof (which minimizes a sum of losses) and the actual FGR algorithm (which uses conditional gradient projection). Please address this discrepancy.
3. Your method's success hinges on the assumption that low-frequency features are more robust and domain-invariant. How does FGR perform on tasks where this assumption is violated, i.e., where high-frequency details are crucial for classification?

I am willing to raise my score if the authors effectively address the issues.

**Limitations:**

yes

**Strengths And Weaknesses:**

Strengths
1. This work proposes a neat and intuitive idea that the combination of frequency-domain low-pass filtering with a projection-based rectification, offering a practical solution for trustworthy model deployment.
2. The method is simple to implement, parameter-light for the crucial trade-off, and delivers consistent gains without target domain access.
3. Experimental validations are sufficient on synthetic and real-world shifts and a semantic shift benchmark.

Weaknesses
1. The novelty needs further clarification. The pipeline is very similar to some existing works, such as Gradient Surgery and Prompt-aligned Gradient, which the authors also mentioned in References. The authors should discuss more explicitly the differences between FGR and these methods to highlight their specific contribution, rather than simply presenting a new application.
2. The model analyzed in the proof of Theorem 1 does not correspond to the actual FGR algorithm. The proof starts by assuming that the FGR model is the minimizer of a joint empirical risk (Eq 11). The gradient update rule for such an objective would be the sum of the two gradients. However, the actual FGR algorithm employs a conditional gradient projection (Eq 8). This gradient surgery is not equivalent to minimizing the sum of two losses.
3. The core assumption of FGR is that distribution shifts primarily alter high-frequency information. This assumption may be invalid or even detrimental for tasks where the critical discriminative information is inherently high-frequency. For example, in fine-grained texture classification or in certain medical imaging applications, low-pass filtering would risk removing the very features essential for classification.

---

> ### Author Rebuttal · Authors · 2026-03-31
>
> We thank the reviewer for the careful reading and for highlighting three central concerns: the novelty claim, the mismatch between the theory and the algorithm, and the scope of the frequency-domain assumption.
>
> ### W1/Q1:Novelty and primary contribution.
>
> We understand the reviewer’s concern about novelty and would like to clarify our contribution precisely. We do **not** claim that DCT-based filtering or gradient projection is individually new. Instead, our main contribution is to identify and address a specific and practically important problem: **target-agnostic calibration under distribution shift**, where target-domain data are unavailable during training.
>
> Within this setting, our key insight is that there is an ID-vs-OOD calibration trade-off: low-pass filtering can improve robustness under shift, but may also hurt ID calibration by removing fine-grained cues. Our solution combines filtering with an asymmetric rectification mechanism. This asymmetry is important: unlike standard gradient-surgery methods that treat objectives symmetrically, our method is designed to preserve progress on the mixed dataset while explicitly preventing a first-order increase in ID calibration loss.
>
> In addition, as mentioned in our response to **Reviewer Feef (W1/Q1)**, we have further supplemented the paper with direct experimental comparisons against PCGrad and CAGrad. These results provide stronger empirical evidence that the proposed asymmetric rectification is better aligned with our objective than symmetric multi-objective gradient combination methods.
>
> ### W2/Q2：Mismatch between the theorem and the actual algorithm
>
> We agree with the reviewer’s criticism. The original theorem presentation was too strong, because it framed FGR as if it were exactly minimizing a scalarized sum of losses, whereas the actual algorithm uses a conditional projection step.
>
> We have revised the theory section so that it matches the implemented optimizer directly. Concretely, the new proposition characterizes FGR as the minimum-perturbation solution of
>
> $$ \min_{\mathbf{g}\in\mathcal{C}} \|\mathbf{g}-\mathbf{g}_{\text{main}}\|_2^2, \qquad \mathcal{C}=\lbrace \mathbf{g} \mid \mathbf{g}^\top \mathbf{g}\_{\text{calib}} \ge 0 \rbrace,$$
>
> that is, the Euclidean projection of the main gradient onto the half-space of directions that do not increase calibration loss to first order. This yields a direct first-order guarantee:
>
> $$
> \mathcal{L}\_{\text{calib}}(\theta-\eta \mathbf{g}_{\text{FGR}})
> \le
> \mathcal{L}\_{\text{calib}}(\theta)+\mathcal{O}(\eta^2)
> $$
>
> for sufficiently small step size $\eta$.
>
> In other words, we fully agree that the earlier joint-risk view should not be presented as an exact equivalence to the algorithm. In the revision, we keep such a view only as a high-level surrogate intuition, while using projection-based proposition as exact theoretical statement.
>
> ### W3/Q3: What if high-frequency information is actually important?
>
> We agree this is an important limitation. Our current target domain is natural-image covariate shift, where high-frequency shortcut cues are often unstable across domains. We do not claim stronger low-pass filtering will always help. In tasks where the discriminative signal is itself high-frequency, aggressive filtering may indeed be harmful.
>
> Our experiments across multiple image benchmarks already demonstrate that the method works robustly over a relatively broad range, including a medical imaging dataset (Camelyon17). To address your concern, we conducted an experiment on the fine-grained bird classification benchmark CUB. The results are shown below:
>
> https://postimg.cc/CnCBbwD4
>
> These results suggest that, on fine-grained tasks, most training-time calibration methods tend to reduce accuracy, likely because such tasks rely heavily on subtle local textures and fine visual details, while calibration-oriented regularization flattens confidence and weakens delicate decision boundaries. In this setting, FGR still achieves the best calibration, while keeping accuracy broadly comparable to other calibration baselines. We agree that on fine-grained tasks, should typically remain the primary concern, and calibration improvement should be considered only after preserving sufficient discriminative performance.
>
> To provide a complementary view, we analyzed ACC and ECE separately across all 15 corruption types on CIFAR-10-C. Because different corruptions perturb different frequencies, the accuracy impact is corruption-dependent: FGR underperforms CE on 5 out of 15 corruptions, but improves calibration on all 15. This is consistent with the design goal of FGR: even when the accuracy trade-off varies across corruption types, the method consistently suppresses overconfidence under shift while preserving accuracy as much as possible.
>
> https://postimg.cc/7bB57P5J
>
> We sincerely hope that our response has addressed and resolved your concerns. We look forward to further communication and  answer any follow-up questions.

---

> > ### Author Rebuttal · Reviewer_Bo8C · 2026-04-03
> >
> > I thank the authors for their thoughtful rebuttal. The proposed revisions have successfully addressed my primary concerns about theoretical soundness. However, as concerns about the degree of novelty and the method's practical trade-offs still linger, I have raised my score to Weak Reject to acknowledge the substantial improvements.

---

> > > ### Author Response · Authors · 2026-04-03
> > >
> > > Thank you again for your thoughtful follow-up and for reconsidering the paper. We greatly appreciate that you found the revised version sufficient to address your primary concern about theoretical soundness, and we are grateful that you raised your score accordingly.
> > >
> > > We also understand that concerns about novelty and practical trade-offs may still remain. Our intended claim is not that the individual components are entirely new, but that FGR formulates target-agnostic calibration under distribution shift in an asymmetric way, which differs from existing gradient-conflict methods. We believe the added comparisons with PCGrad and CAGrad further clarify this point.
> > >
> > > We fully respect your judgment, and we sincerely appreciate your constructive feedback. If you feel that these clarifications and additional experiments better distinguish our work from prior methods, we would be very grateful if you could also reconsider the **originality** assessment.

---

### Official Review · Reviewer_Feef · 2026-03-11

**Soundness:** 3
**Presentation:** 3
**Significance:** 3
**Originality:** 3
**Overall Recommendation:** 4
**Confidence:** 3

**Summary:**

This paper proposes FGR, a target-agnostic training framework for robust calibration under distribution shift. It combines DCT-based low-pass filtering (to suppress high-frequency shortcuts and encourage domain-invariant features) with gradient rectification (treating ID calibration as a hard constraint via geometric projection when gradients conflict).

**Compliance With Llm Reviewing Policy:**

Affirmed.

**Final Justification:**

Thank you for the detailed rebuttal. The authors have addressed most of my concerns satisfactorily, and I appreciate the additional experiments and clarifications. I maintain my current score.

**Key Questions For Authors:**

1. How does FGR compare to existing multi-objective gradient methods (PCGrad, CAGrad)?
2. What is the joint sensitivity of ρ and λ?
3. What is the principled justification for starting filtering at epoch 200?

**Limitations:**

The near-zero accuracy improvement on semantic shift (Office-Home) should be discussed more candidly as a limitation of the frequency-based approach.

**Strengths And Weaknesses:**

Strengths
1. Exceptionally clean ablation — Table 3 clearly isolates each component's contribution and limitation
2. Parameter-free gradient projection is elegant and practical
3. Theorem 1 provides non-trivial generalization bounds
4. Broad experimental coverage: synthetic corruption + real-world shift + semantic shift
5. Compatibility with post-hoc methods verified

Weakness
1. [Major] Individual components are from prior work: DCT filtering (Geirhos et al., Fridovich-Keil et al.) and gradient projection for conflicting objectives (PCGrad, Yu et al.). The combination is clean but fundamentally incremental.
2. [Minor] Joint sensitivity analysis of λ and ρ is missing
3. [Minor] The two-stage strategy (start filtering at epoch 200) lacks principled justification
4. [Minor] On Office-Home (Table 2), accuracy is nearly identical to CE baseline (63.19 vs 63.22) — benefit is limited to calibration under semantic shift

---

> ### Author Rebuttal · Authors · 2026-03-31
>
> We thank the reviewer for the positive comments on the ablations, coverage, and practical value, and for the concrete suggestions regarding novelty, sensitivity analysis, the warm-up strategy, and the Office-Home result.
>
> ### W1/Q1: Novelty and comparison to existing multi-objective gradient methods
>
> We agree that the individual ingredients of FGR are related to prior work. Following your suggestion in Q1, we further compared FGR against two strong multi-objective gradient baselines, **PCGrad** and **CAGrad**, by replacing the rectification rule in FGR with their respective gradient-combination mechanisms. The results on two real-world datasets are shown below.
>
> Due to space limits, we provide experimental results via an anonymous link for reference: https://postimg.cc/svhmMWvn
>
> These results show that **PCGrad and CAGrad are indeed competitive baselines**, and we appreciate the reviewer for prompting this comparison. At the same time, the results also support our intended point: our **asymmetric rectification** is more aligned with the goal of FGR. We are **not** seeking a symmetric compromise between two peer tasks. Instead, we aim to optimize robustness on the mixed dataset $\mathcal{D}_{\mathrm{mix}}$ while explicitly preventing a first-order increase in ID calibration loss. This objective geometry is different from PCGrad/CAGrad, and we believe the revised ablation will make that distinction clearer. We will include PCGrad and CAGrad as stronger baselines in the revision.
>
> ### W2/Q2:  Joint sensitivity analysis of λ and ρ
>
> We thank the reviewer for this suggestion. The current appendix indeed only reports **one-dimensional sensitivity analyses** for $\lambda$ and $\rho$. We have now conducted a small-scale **two-dimensional sensitivity analysis** on CIFAR-10/100 and the real-world shifted dataset iWildCam, using $\rho \in \{0.03, 0.05, 0.1\}$ and $\lambda \in \{15,18,25\}$.
>
> ACC and ECE heatmaps within the distribution (CIFAR-10, CIFAR-100): https://postimg.cc/TL4ZtQ3R
>
> ACC and ECE heatmaps for data points outside the distribution (Cifar10-C, Cifar100-C, iWildCam): https://postimg.cc/cgbVFTLJ
>
> The main trend is consistent with our earlier 1D analysis: **FGR is not overly sensitive to $\lambda$ within a moderate range**, while the improvement in OOD calibration is controlled more strongly by $\rho$. At the same time, there is a clear trade-off between ID and OOD calibration: overly strong filtering leads to a sharp drop in ID calibration performance. We will include the complete 2D sensitivity analysis in the revision.
>
> ### W3/Q3:  Why starting filtering at epoch 200?
>
> We agree that the previous manuscript did not sufficiently justify this design. This is indeed an **empirically motivated warm-up strategy**. Our observation is that introducing filtering and rectification too early can disrupt initial decision-boundary formation and lead to unstable training. We therefore first allow the model to learn basic discriminative structure using the main loss, and only then activate FGR. To support this point, we conducted a start-epoch ablation.
>
> Due to space limits, we provide experimental results via an anonymous link for reference: https://postimg.cc/bZPxdF7F
>
> These results support our warm-up design: activating FGR too early is not reliably beneficial, whereas a later start yields a more stable balance. We will present this more clearly as an **empirical design choice**.
>
> ### W4/Limitations: Accuracy performance on Office-Home
>
> We agree and will discuss this more candidly. As shown in Table 2 of the current manuscript, the accuracy difference between CE and FGR on Office-Home is indeed negligible, while the main gain lies in calibration. Our purpose in including this experiment was not to claim a strong accuracy improvement, but rather to show that the method is **not limited to simple corruption-style shifts**, and can also improve calibration under **semantic shift**. We will revise the wording accordingly to better reflect this point.
>
> We thank the reviewer again for the concrete and highly actionable suggestions. The additional comparisons and sensitivity studies have materially improved the paper, and we will integrate them into the revision. We hope our response could address your concerns, and we also welcome any further feedback you might provide to enhance our manuscript.

---

> > ### Author Rebuttal · Reviewer_Feef · 2026-04-03
> >
> > I thank the authors for the thorough rebuttal. The additional PCGrad/CAGrad comparison, joint sensitivity analysis, and start-epoch ablation directly address my concerns.
> > For the revision:
> >
> > Include PCGrad/CAGrad in the main comparison tables, and explicitly discuss the asymmetric-vs-symmetric design distinction in the method or related work section.
> > Present the 2D (λ, ρ) sensitivity heatmaps as proper figures in the appendix, with a brief note on the ID–OOD trade-off surface.
> > Add the start-epoch ablation to the appendix and acknowledge the warm-up as an empirical heuristic in the main text.
> > Discuss the Office-Home accuracy limitation more candidly — frequency filtering is better suited for low-level texture shortcuts than high-level semantic gaps, and this scope should be stated explicitly.
> > The proof's final step (Eq. 35 → Eq. 12) absorbs constant factors somewhat loosely. A brief remark on when the bound is expected to be tight or loose would help.

---

> > > ### Author Response · Authors · 2026-04-04
> > >
> > > Thank you very much for the thoughtful follow-up and for the clear revision guidance. We appreciate these concrete suggestions, and we will revise the paper accordingly: we will include PCGrad/CAGrad in the main comparison tables, clarify the asymmetric-versus-symmetric design distinction more explicitly, add the 𝜆,𝜌 sensitivity heatmaps and the start-epoch ablation to the appendix, describe the warm-up as an empirical heuristic in the main text, and discuss the Office-Home accuracy limitation more candidly to better state the scope of the method.
> > >
> > > We have also re-rendered the 𝜆,𝜌 sensitivity results over a larger hyperparameter range as formal heatmaps. The updated images are as follows:
> > >
> > > In-Distribution ACC and ECE heatmaps (CIFAR-10, CIFAR-100): https://postimg.cc/WtqtdDDK
> > >
> > > OOD ACC and ECE heatmaps (Cifar10-C, Cifar100-C, iWildCam): https://postimg.cc/qg6gNthV
> > >
> > > Regarding the proof detail you pointed out, thank you for the careful reading. We agree that the final step from Eq. 35 to Eq. 12 currently absorbs constant factors somewhat loosely for presentation simplicity. In the revision, we will make this explicit and add a brief remark on when the bound is expected to be tighter or looser. More specifically, the bound is expected to be tighter when the controlled filtering induces only a small discrepancy between $D_{mix}$ and $D_{orig}$, so that the discrepancy term remains relatively small. It may become looser when the filtering is more aggressive, or in regimes with limited samples and relatively high model complexity.
> > >
> > > Thank you again for the highly actionable feedback. We hope these clarifications and additions help resolve the remaining concerns, and we would sincerely appreciate any updated assessment you may consider appropriate.

---

### Official Review · Reviewer_9VSC · 2026-03-12

**Soundness:** 4
**Presentation:** 4
**Significance:** 3
**Originality:** 2
**Overall Recommendation:** 4
**Confidence:** 5

**Summary:**

This paper proposes Frequency-aware Gradient Rectification (FGR), a training-time framework to improve model calibration under distribution shift without requiring target domain data. FGR applies DCT-based low-pass filtering to a subset of training images to suppress high-frequency shortcut cues and employs gradient rectification that projects the main loss gradient orthogonally to the calibration loss gradient whenever the two conflict. Experiments on synthetic and real world distribution shift benchmarks show consistent calibration improvements on target set.

**Compliance With Llm Reviewing Policy:**

Affirmed.

**Final Justification:**

Thank you for the thorough rebuttal. The authors have addressed my main concerns satisfactorily, so I am updating my score to **Weak Accept**. I am not rating the paper higher because I still find the overall novelty somewhat limited, with the main contribution lying in the formulation and combination of existing ideas, and because the proposed DCT-based solution appears restricted to image-domain settings, which limits its broader applicability.

**Key Questions For Authors:**

1. The complementary nature of filtering-for-OOD and rectification-for-ID calibration appears to be one of the paper's most insightful findings. Why has this not been foregrounded as a primary contribution addressing the ID vs. OOD calibration Pareto frontier, rather than focusing predominantly on OOD calibration?
2. Given these complementary benefits between the two mechanisms, can you provide practical "when to use what" deployment guidance to further strengthen the paper's utility for practitioners?
3. Table 3 excellently demonstrates the complementary benefits of your components, but it is restricted entirely to synthetic shifts. Can you provide a similar ablation study for a real-world distribution shift benchmark (e.g., iWildCam or Camelyon17) to confirm that these dynamics hold beyond synthetic noise?
4. Could you clarify the "Rectification Only" baseline in Table 3 regarding what it measures when gradients are computed solely on clean images? Specifically, why do these gradients conflict on clean data, and what exactly does the projection resolve to improve ID calibration?
5. Given the goal of a completely target-agnostic method, can you explicitly clarify how hyperparameters were tuned for both the baselines and FGR? Specifically, were any target OOD samples used during this tuning process, or was it strictly isolated to the ID validation set?


*References*

1. Yang and Soatto, *FDA: Fourier Domain Adaptation for Semantic Segmentation*, CVPR 2020.
2. Lin et al., *Deep Frequency Filtering for Domain Generalization*, CVPR 2023.
3. Yu et al., *Gradient Surgery for Multi-Task Learning*, Neurips 2020.
4. Lin et al., *Uncertainty Weighted Gradients for Model Calibration*, CVPR 2025.
5. Sahoo et al., *A Layer Selection Approach to Test Time Adaptation,* AAAI 2025.

**Limitations:**

yes

**Strengths And Weaknesses:**

*Strengths*
- The proposed simple modification to the training pipeline is highly impactful for rapid adoption by subsequent works, improving calibration with minimal reduction in overall predictive performance.
- Extensive experimental evaluation across synthetic corruptions, real-world distribution shifts, and semantic shifts demonstrates consistent and significant improvements in OOD calibration scores.
- The clean ablation results effectively highlight how the low-pass filtering and gradient rectification components provide genuinely complementary benefits for OOD and ID calibration, respectively.


*Weaknesses*
- The proposed DCT-based filtering mechanism is fundamentally image-specific, limiting its overall generality and applicability for other modalities such as text or structured data where no natural analog exists.
- The approach possesses limited novelty as it combines two well-established ideas applied to a new calibration task: frequency filtering for OOD robustness [1, 2], and gradient projection for conflict resolution [3, 4, 5].
- The "parameter-free" framing is somewhat misleading because the proposed method still requires tuning key hyperparameters (such as the focusing parameter, filtering ratio, and start epoch) per dataset and architecture, comparable to the evaluated baselines.

---

> ### Author Rebuttal · Authors · 2026-03-31
>
> We thank the reviewer for the thoughtful feedback, especially for recognizing the complementary roles of filtering and rectification and for highlighting the practitioner-facing value of the paper. We respond to the weaknesses and questions point by point below.
>
> ### W1: Applicability to other modalities
>
> We agree that the current FGR framework is image-centric, since the filtering mechanism is defined in the visual frequency domain. We will clarify this scope more explicitly in the revision. Accordingly, our claim is not modality-universal calibration, but rather a target-agnostic calibration framework for image-domain distribution shift. Our broad image-benchmark results support its effectiveness in this setting, and extending the gradient-rectification idea to other modalities is an important direction for future work.
>
> ### W2/Q1: Novelty and primary contribution.
>
> We understand the reviewer’s concern and do not claim that either DCT-based filtering or gradient projection is individually new. Our main contribution is to identify a practically important problem—target-agnostic calibration under distribution shift—and to formulate it in an asymmetric way. Unlike existing gradient-conflict methods that treat multiple objectives more symmetrically, FGR does not optimize robustness and ID calibration as two peer goals. Instead, it improves robustness on $\mathcal{D}_{\mathrm{mix}}$   while using rectification only to prevent a first-order increase in ID calibration loss. We will revise the corresponding statements to make this point much clearer.
>
> ### W3: Not entirely parameter-free.
>
> We appreciate this observation and agree that our wording should be more precise. By “parameter-free,” we only meant that the **rectification step introduces no additional loss-balancing weight**. FGR still requires tuning the focusing parameter and filtering-related hyperparameters. Empirically, however, $\rho$ and $\lambda$ are relatively robust across datasets, while $\gamma$ is the main parameter that benefits from careful tuning. We will replace “parameter-free” with more accurate wording such as **“without introducing an additional loss-balancing weight.”**
>
> ### Q2: Practical guidance on when to use which mechanism
>
> We agree that a short practitioner-oriented discussion would strengthen the paper. In brief, **FGR is most suitable for tasks that are not highly fine-grained and may face data-distribution changes during collection or deployment**, such as changes in weather, illumination, or imaging conditions. In such setup, improving OOD calibration is especially important, and our method is designed for exactly this scenario.
>
> Practically, **filtering** helps suppress unstable high-frequency shortcut cues, while **rectification** helps preserve ID calibration. We also observe that a **larger filtering ratio $\rho$** usually leads to **better OOD calibration**, although excessively large $\rho$ may introduce a stronger trade-off with ID performance.
>
> ### Q3: Real-world ablation.
>
> Thank you for this very useful suggestion. We re-ran a real-world ablation on Camelyon17 and iWildCam under a unified training protocol and hyperparameter setting ($\gamma=5$, $\rho=0.05$).
>
> https://postimg.cc/jwSZgdtQ
>
> These results support the same qualitative conclusion as our synthetic ablation: the two components behave differently, and the full method achieves the most favorable overall balance. On Camelyon17, FGR gives the best validation ECE, test accuracy, and test ECE. On iWildCam, Filter Only is already strong in shifted ECE, but FGR achieves the best overall combination of shifted accuracy and NLL while remaining competitive in ECE. We will add this ablation to the revision.
>
> ### Q4: Clarification of “Rectification Only”.
>
> This is an important question. Even on clean inputs, the main objective and the calibration objective are not fully aligned. The main loss still tends to sharpen predictive distributions in order to improve classification, whereas the calibration loss penalizes mismatch between confidence and empirical correctness. Therefore, gradient conflict can arise even when both gradients are computed from clean data. In the revision, we will clarify this point more explicitly.
>
> ### Q5: Hyperparameter tuning process
>
> We fully agree that this must be stated explicitly. In our setup, **all methods are tuned using ID validation data only**. For FGR, as reported in Appendix C.4.2, we generally fix the filtering parameters to $\rho = 0.05$ and sample $\lambda$ from $\{15,18,25\}$ for each image, then tune the focusing parameter $\gamma$ based on the best calibration performance on the **ID validation set**. The final OOD performance is evaluated only once on the shifted test data. We will move this clarification into the main paper to avoid any ambiguity.
>
> We hope that our response has addressed and resolved your concerns. We look forward to further communication and are happy to answer any follow-up questions.

---

> > ### Author Rebuttal · Reviewer_9VSC · 2026-04-03
> >
> > Thank you for the thorough rebuttal. The authors have addressed my main concerns satisfactorily, so I am updating my score to **Weak Accept**. I am not rating the paper higher because I still find the overall novelty somewhat limited, with the main contribution lying in the formulation and combination of existing ideas, and because the proposed DCT-based solution appears restricted to image-domain settings, which limits its broader applicability.

---

> > > ### Author Response · Authors · 2026-04-07
> > >
> > > Thank you again for the thoughtful follow-up and for updating your score! We sincerely appreciate that you found our rebuttal sufficient to address your main concerns.
> > >
> > > We understand your remaining reservations regarding novelty and image-domain scope. As we clarified earlier, our intended claim is not that the individual ingredients are entirely new, but that FGR formulates target-agnostic calibration under distribution shift in an asymmetric way, rather than treating robustness and ID calibration as two peer objectives.
> > >
> > > In addition, following Reviewer Feef’s W1/Q1 suggestion, we further compared FGR against two strong multi-objective gradient baselines, PCGrad and CAGrad, by replacing the rectification rule in FGR with their respective gradient-combination mechanisms. As shown in the table below, these methods are indeed competitive baselines, but the results also support our intended point that FGR is better aligned with our objective of improving robustness on the mixed dataset while preventing a first-order increase in ID calibration loss. We will include these comparisons in the revised paper.
> > >
> > > * Performance on the ID validation set and OOD test set for two real-world datasets.
> > >
> > > | Camelyon17 | Val Acc   | Val ECE    | Test Acc  | Test ECE   | Test NLL   |
> > > |-|-|-|-|-|-|
> > > | PCGrad     | 90.56     | 0.0253     | **88.68** | 0.0338     | 0.2909     |
> > > | CAGrad     | 90.34     | 0.0172     | 87.10     | 0.0128     | 0.3165     |
> > > | FGR        | **90.90** | **0.0092** | 87.37     | **0.0114** | **0.3091** |
> > >
> > > | iWildCam | Val Acc   | Val ECE    | Test Acc  | Test ECE   | Test NLL   |
> > > |-|-|-|-|-|-|
> > > | PCGrad   | 59.94     | 0.1051     | **77.21** | 0.0274     | 1.0835     |
> > > | CAGrad   | 60.39     | 0.0948     | 74.92     | 0.0451     | 1.1044     |
> > > | FGR      | **60.61** | **0.0819** | 76.22     | **0.0262** | **1.0597** |
> > >
> > > We also agree that the current DCT-based filtering is image-specific. More generally, however, we view the robustness-inducing transformation as modality-dependent, while the gradient-rectification principle itself is broader and could potentially be combined with analogous transformations in other domains in our future work.
> > >
> > > Thank you again for your constructive feedback.

---

### Official Review · Reviewer_F7DM · 2026-03-23

**Soundness:** 3
**Presentation:** 3
**Significance:** 2
**Originality:** 3
**Overall Recommendation:** 5
**Confidence:** 3

**Summary:**

This paper studies robust calibration under distribution shifts which occur often at deployment time. It proposes Frequency-aware Gradient Rectification (FGR), a task-agnostic calibration method during training to ensures calibrated models even when the distribution of data it is applied on differ from the training distribution. The idea is to apply a low-pass filtering on a subset of the training images to enforce the model to focus on domain-agnostic features. To mitigate the degradation of the calibration in-distribution of this approach, a gradient rectification is applied to ensure that the parameters update does not degrade the calibration on images following the training distribution. The benefits of this approach is shown on experiments with ResNet and DenseNet models on synthetic and real-world data, showcasing the superiority of FGR for calibration both in ID domain and under distribution shifts.

**Compliance With Llm Reviewing Policy:**

Affirmed.

**Final Justification:**

I thank the authors for a comprehensive rebuttal that addressed all my concerns. I updated my score from 4 (weak accept) to 5 (accept) to reflect the improvement. The 18% overhead of the method might limit the applicability of the method at very large scales, but I believe the overall contribution is still valuable for small/medium datasets.

**Key Questions For Authors:**

- Regarding the theory, I wonder how much the filtering impacts the Wasserstein distance between $D_{ori}$ and $D_{mix}$. Could the authors show the evolution with less and less controlled filtering on this distance?
- Related to the weaknesses, could the authors discuss the applicability of their approach on ViT models (or eventually run sanity check experiments). I wonder whether the low-pass filtering will still be beneficial. Moreover, recent large foundations models tend to be more calibrated than previous ResNets models (see [1, 2]), which would again be an interesting comparison to see the versatility of the proposed method.

[1] Guo et al. On Calibration of Modern Neural Networks. In ICML 2017

[2] Minderer et al. Revisiting the Calibration of Modern Neural Networks. In NeurIPS 2020

**Limitations:**

Yes, although it is mainly discussed in the end of the appendix. Having more discussion in the main paper would be better. Indeed, showing that the overhead over the usual CE training is small improves the strength of the approach and showing that FGR-Scratch adds a non negligible training overhead is important for transparency (notably given the fact that it remains competitive with other calibration methods).

**Strengths And Weaknesses:**

**Strengths**
- The paper is well presented with clear method, results and analysis
- The experimental claims in terms of calibration are supported with detailed experiments, sensitivity analysis and ablation studies
- The choice of baselines seem sound and the qualitative benefits of FGR is supported adequately
- The method design is well supported with previous literature and intuitive explanations

**Weaknesses**

I list below what I believe are weaknesses but I would be happy to be corrected if I misunderstood some parts of the work.
- The current results are presented without seeds which makes it hard to know whether the differences are statistically different. Given the fact that this paper proposes an empirical methodology, I believe this is of great importance
- The practical applicability of the approach might be limited given the computational overhead of 18% of the proposed method. Although it remains comparable to other calibration approaches, this becomes prohibitive when the size of training datasets increases as is the case with large vision foundation models (and all the more with other large models like LLMs).
- Although the experiments are done on a variety of datasets, all the models are convolutional models. While the limitation on image datasets is not a weakness in my opinion, I believe more diversity on the models would improve the current submission. I particular, CNNs tend to create local filters of the images, which make intuitive the benefits of the low-pass filtering in the frequency domain. It would be interesting to know whether this is still the case with ViT that create patches of data. Given that foundation models (where pretraining is prohibitive and for which ensuring calibration at test time under distribution shift is of great importance) are increasingly based on transformers models, such comparison would increase the applicability of the proposed method.

Overall, the paper is interesting and the proposed approach seem promising. The current evaluation of ResNets models limit the applicability and impact of the approach which justify my score. I remain open to improve it depending on the authors rebuttal.

---
**Updated score: from 4 to 5**

---

> ### Author Rebuttal · Authors · 2026-03-31
>
> We thank the reviewer for the positive assessment of the presentation, empirical study, and qualitative analysis, and for the concrete suggestions on reproducibility, efficiency, and architecture diversity. In response, we conducted additional experiments on (i) the effect of filtering on the Wasserstein distance between $D_{\mathrm{orig}}$ and $D_{\mathrm{mix}}$, and (ii) ViT-style architectures. We respond point by point below.
>
> ### W1: Results are presented without seeds.
>
> We appreciate this concern and fully agree that statistical transparency is important, especially for an empirical paper. We apologize that the original manuscript did not explicitly report dispersion statistics in the main paper due to space constraints. As stated in Appendix C.1, all reported results were already averaged over three independent runs with different random seeds.
>
> To further address your concern, we additionally re-ran the main comparisons on CIFAR-10/100 using the fixed random seeds $\{1,2,3\}$, and now report the corresponding **mean $\pm$ standard deviation**. We also conducted Welch's t-test, we find that the main statistically reliable gain of FGR is on OOD calibration. In both CIFAR-10-C and CIFAR-100-C, FGR significantly improves ECE over both CE and DFL.
>
> Due to space limits, we provide experimental results via an anonymous link for reference.
> https://postimg.cc/HVf0mNLN
>
> These results continue to support the same conclusion as in the paper: while FGR remains competitive on clean data, it provides the clearest advantage under distribution shift, particularly in shifted ECE.
>
> ### W2/Limitations: Practical applicability.
>
> We appreciate this concern and agree that transparency about training cost is important. The current manuscript already reports that training-from-scratch incurs about **18% additional overhead** relative to standard training, and also includes a **two-stage fine-tuning** strategy intended to reduce cost. In the revision, we will move this discussion into the main paper, make the cost/benefit trade-off more explicit, and highlight that the lighter fine-tuning variant achieves comparable calibration performance with substantially lower additional cost when a base CE model is already available.
>
> ### W3/Q2: All evaluated models are convolutional; what about ViT?
>
> Thank you for this constructive suggestion. We have now conducted ViT-based sanity checks on both synthetic and real-world benchmarks. The results are summarized below:
>
> - ViT-small on CIFAR-10 / CIFAR-10-C: https://postimg.cc/HjMsKvfj
>
> - DeiT-small on CIFAR-10 / CIFAR-10-C: https://postimg.cc/87fSJ7NC
>
> - DeiT-small on Camelyon17: https://postimg.cc/Yj2BNnt7
>
> These results suggest that the benefit of FGR is **not tightly tied to convolution-specific local filtering behavior**. Even on ViT-style architectures, FGR improves both clean calibration and calibration under distribution shift, while often also improving shifted accuracy. In the revision, we will include these results as preliminary evidence that the method is compatible with transformer-based image models as well.
>
> ### Q1: How does filtering affect the Wasserstein distance?
>
> Thank you for the suggestion. We have now added an empirical analysis of the feature-space Wasserstein distance between $D_{\mathrm{orig}}$ and $D_{\mathrm{mix}}$ under different filtering strengths and filtering ratios. We compute the Gaussian 2-Wasserstein distance on penultimate-layer features from a ResNet-50 CE model on CIFAR-10, using 4096 training samples.
>
> * Fixed $\rho = 0.1$, varying $\lambda$:
>
> | $\lambda$           |    25 |    18 |    15 |    10 |     7 |
> | ------------------- | ----: | ----: | ----: | ----: | ----: |
> | Feature-space $W_2$ | 0.517 | 0.522 | 0.510 | 0.457 | 0.471 |
>
> * Fixed $\lambda = 15$, varying $\rho$
>
> | $\rho$              |  0.05 |  0.10 |  0.20 |  0.40 |
> | ------------------- | ----: | ----: | ----: | ----: |
> | Feature-space $W_2$ | 0.300 | 0.510 | 1.018 | 1.952 |
>
> The main empirical takeaway is that the perturbation strength is controlled **primarily by $\rho$**: increasing the fraction of filtered samples leads to a clear increase in the feature-space distribution gap. In contrast, varying $\lambda$ within a moderate range has a noticeably milder effect. This is also consistent with our joint ablation results, where $\rho$ acts as the main trade-off factor, while $\lambda$ is relatively robust within a reasonable range.
>
> We will incorporate the additional ViT and Wasserstein analyses into the revision and release the code upon publication. We hope our response could address your concerns, and we also welcome any further feedback you might provide to enhance our manuscript.

---

> > ### Author Rebuttal · Reviewer_F7DM · 2026-04-01
> >
> > I thank the authors for a comprehensive rebuttal. Each point of my review is addressed. The 18% overhead of the method might still limit its applicability at very large scales. Still, I believe this does not hinder its contribution, given its clear scope for small/medium datasets. I updated my score accordingly from 4 (weak accept) to 5 (accept).
> >
> > Best,
> >
> > Reviewer F7DM

---

> > > ### Author Response · Authors · 2026-04-01
> > >
> > > Thank you very much for your careful reading and for the encouraging update. We appreciate your recognition of both the contribution and its practical scope. In the revision, we will further evaluate the overhead of our method on larger-scale datasets and provide clearer practical guidance on when and how it can be used effectively.
> > >
> > > Best regards,
> > > The Authors

---

### Decision · Program_Chairs · 2026-04-30

**Decision:**

Accept (regular)

**Comment:**

The paper presents a DCT-based frequency based filtering mechanism and a gradient rectification technique (FGR) to handle the challenge of calibration in in-domain and out-domain scenarios.

Some core strengths of the submission include:

- the paper is well-presented and clearly written
- proposed technique allows simple modification to baseline
- extensive results are presented covering various shifts and corruption types
- presents a neat and intuitive idea and is useful solution for trustworthy model deployment

Among the important concerns highlighted by the reviewers include:
- computational overhead related to other methods at training time
- results limited to CNN-based architecture
- DCT-based filtering mechanism is fundamentally image-specific
- novelty concerns as it appears to be a combination of known components in literature
- the assumption of domain-specific information lying in high frequency signals is restrictive

In the post-rebuttal phase, three reviewers (F7DM, 9VSC, Feef) mentioned that their concerns have been resolved satisfactorily and either kept their score or increased ratings with final ratings to be an accept and two weak accepts. Reviewer Bo8C posted that his/her primary concerns about theoretical soundness have been addressed, however, the concerns on degree of novelty and method's practical trade-offs remains.
AC thinks that, the concern on technical novelty can be discounted, as also acknowledged by Feef reviewer in the post-rebuttal. The proposed FGR realizes target-agnostic calibration under distribution shift in an asymmetric way, which differs from existing gradient-conflict methods and further includes relevant comparisons with PCGrad and CAGrad to strengthen this point. In addition, all reviewers agree that the method is simple to implement, the paper is well-written, and results show effectiveness on several datasets. Therefore, AC decides to recommend acceptance and recommends authors to include important result in the final revision.